# Structural insights into RNA bridging between HIV-1 Vif and antiviral factor APOBEC3G

Takahide Kouno [1] ✉, Satoshi Shibata [1,5], Megumi Shigematsu [2], Jaekyung Hyun [1,6], Tae Gyun Kim[1,7], Hiroshi Matsuo [3] & Matthias Wolf [1,4] ✉

Great effort has been devoted to discovering the basis of A3G-Vif interaction, the key event of HIV's counteraction mechanism to evade antiviral innate immune response. Here we show reconstitution of the A3G-Vif complex and subsequent A3G ubiquitination in vitro and report the cryo-EM structure of the A3G-Vif complex at 2.8 Å resolution using solubility-enhanced variants of A3G and Vif. We present an atomic model of the A3G-Vif interface, which assembles via known amino acid determinants. This assembly is not achieved by protein-protein interaction alone, but also involves RNA. The cryo-EM structure and in vitro ubiquitination assays identify an adenine/guanine base preference for the interaction and a unique Vif-ribose contact. This establishes the biological significance of an RNA ligand. Further assessment of interactions between A3G, Vif, and RNA ligands show that the A3G-Vif assembly and subsequent ubiquitination can be controlled by amino acid mutations at the interface or by polynucleotide modification, suggesting that a specific chemical moiety would be a promising pharmacophore to inhibit the A3G-Vif interaction.

HIV-1 hijacks a host ubiquitin ligase complex using a viral infectivity factor, Vif, to degrade the antiviral factor, APOBEC3G (A3G)[1–5]. A3G is a human, single-stranded DNA (ssDNA) deaminase that catalyzes fatal mutations in genomes of viruses such as HIV-1 [Reviewed in [6–9]]. Since its discovery in 2002[1], extensive research has been conducted globally to explore A3G functions and to overcome the HIV-1 pandemic by understanding its intrinsic antiviral mechanism. Expressed A3G binds to RNAs to form high-molecular-mass (HMM) protein-RNA complexes in the cytoplasm[10,11]. Upon infection by *vif*-deficient HIV-1, host A3G binds to viral genomic and nonviral RNAs and is incorporated into budding virions, together with viral components. Encapsidated A3G recognizes the natal ssDNA reverse transcript of the viral genome in capsids, and catalyzes cytosine deamination, i.e., 2′-deoxycytosine to 2′-deoxyuracil conversions in the ssDNA, impairing viral amplification[12–15]. Recent studies found that the viral genome is protected by the capsid assembly from viral entry until nuclear import[16,17], indicating that A3G expressed in infected cells is excluded from invading viral components, including viral RNA and DNA. Therefore, encapsidation of A3G is the only way to access viral ssDNA. HIV-1 viral infectivity factor (Vif) protein, however, disturbs A3G encapsidation (Fig. 1a)[1,3–5]. Vif hijacks host core binding factor β (CBFβ) to form a stable Vif-CBFβ complex, which associates A3G with the CUL5 ubiquitin ligase complex[18,19]. The captured A3G is ubiquitinated and then degraded in proteasomes before its

[1]Molecular Cryo-Electron Microscopy Unit, Okinawa Institute of Science and Technology Graduate University, 1919-1 Tancha, Onna-son, Okinawa 904-0495, Japan. [2]Computational Medicine Center, Sidney Kimmel Medical College, Thomas Jefferson University, Philadelphia, PA 19107, USA. [3]Cancer Innovation Laboratory, Frederick National Laboratory for Cancer Research, Frederick, MD 21702, USA. [4]Institute of Biological Chemistry, Academia Sinica, 128 Academia Road Sec. 2, 115 Taipei, Taiwan. [5]Present address: Division of Bacteriology, Department of Microbiology and Immunology, Faculty of Medicine, Tottori University, 86 Nishi-cho, Yonago-shi, Tottori 683-8503, Japan. [6]Present address: School of Pharmacy, Sungkyunkwan University, Suwon-si, Gyeonggi-do 16419, Republic of Korea. [7]Present address: Department of Efficacy Evaluation, Innovation Center for Vaccine Industry, Gyeongbuk Institute for Bio Industry, Gyeongsanbuk-do 36618, Republic of Korea. ✉e-mail: 1st.soluble.fla3g@gmail.com; matthias.wolf@oist.jp

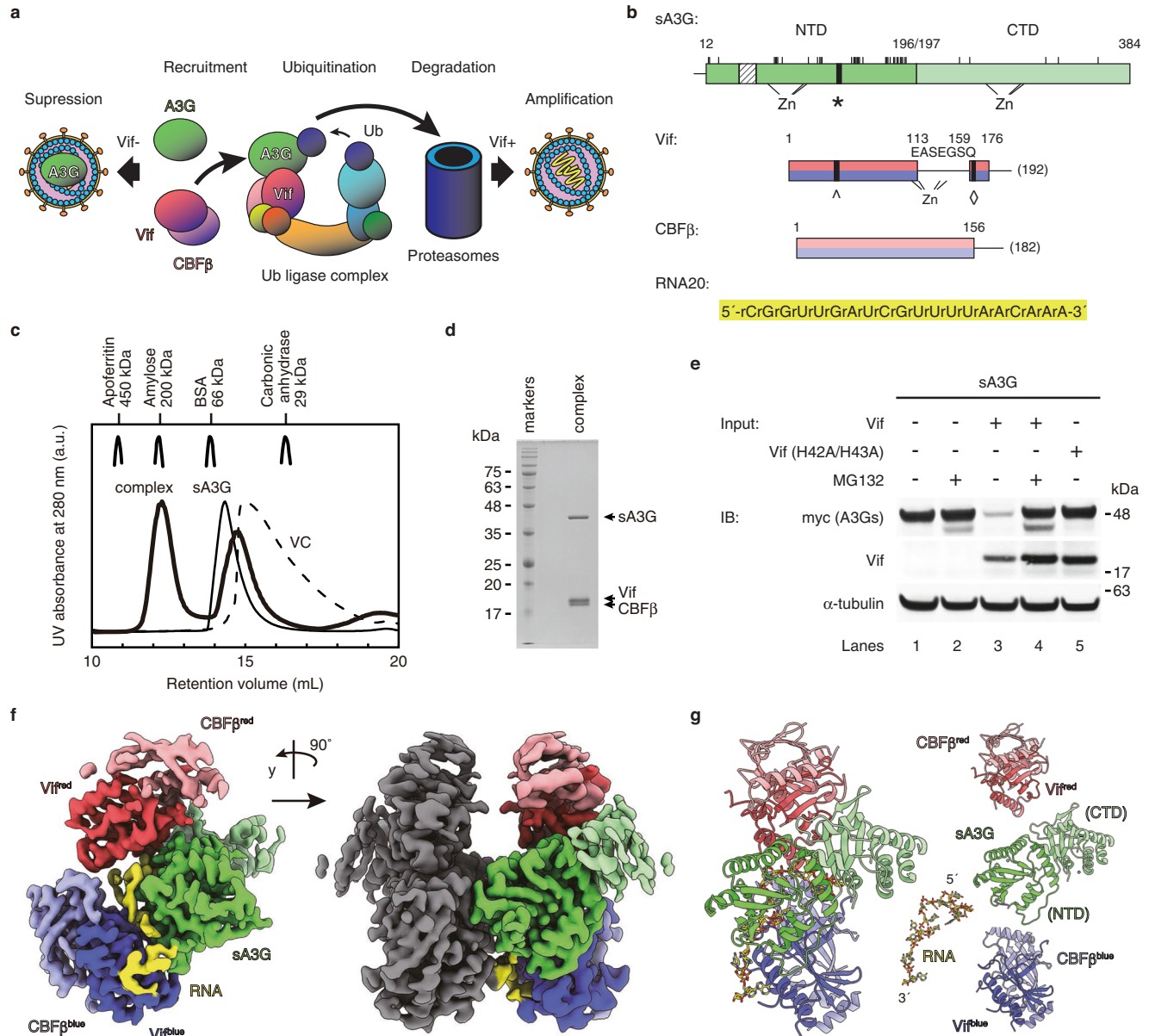

**Fig. 1 | Protein preparation and cryo-EM reconstruction of the sA3G-VC-RNA20 complex. a** Schematic of Vif counteraction mechanism against A3G in an infected cell. Only encapsidated A3G restricts HIV-1 infection (Suppression). Vif recruits A3G into the ubiquitin ligase complex to eliminate it from the cytoplasm, which produces 'healthy' virions (Amplification). **b** Protein and RNA constructs used in this study include sA3G NTD (green) and CTD (light green), Vif (red and blue), and CBFβ (light red and light blue). Truncations are indicated by horizontal lines, and the peptide EASEGSQ was inserted between amino acids 113 and 159 of Vif. In the scheme of sA3G, amino acid replacements are represented by vertical lines and a hatched box (amino acid sequence is provided in Supplementary Fig. 1e). The consensus Vif-binding site, 126-FWDPD-130, is indicated by a filled box with an asterisk. In the Vif construct, filled boxes labeled with a hat (ˆ) and diamond (◊) depict the consensus A3G-binding site, 42-HHY-44, and PPLP motif, respectively. Positions of zinc binding motifs are marked beneath the rectangles. Numbers indicate amino acid residues located at domain boundaries or the C-terminal

residue of wild-type protein (in parentheses). **c** Size exclusion chromatograms of sA3G (thin line), VC (dashed), and sA3G-VC-RNA20 complex (bold). Molecular weight standards are shown for comparison. **d** SDS-PAGE profile of purified sA3G-VC-RNA20 complex (*n* = 1). **e** Vif-induced degradation of sA3G in human cells. While sA3G expression was attenuated in the presence of Vif (lane 3), it was recovered by addition of proteasome inhibitor MG132 (lane 4) or Vif-deficient mutation, H42A/ H43A (lane 5). The assay was independently performed 3 times (*n* = 3) for each condition. The figure is a composite of Supplementary Fig. 1f. Source data are provided as a Source data file. **f** Cryo-EM reconstruction of the sA3G-VC-RNA20 complex at 2.8 Å resolution, contoured at a threshold of 6σ. The complex is a C2-symmetric dimer (right). The asymmetric unit (left) is colored in the same manner as in (**b**). **g** Ribbon presentation of the atomic model of sA3G-VC-RNA20 complex and dissection of its components. The orientation and coloring scheme follow the right panel in (**f**).

encapsidation. HIV-1 harboring a functional *vif* gene, thereby produces 'healthy' virions without A3G, advancing the viral life cycle, i.e., successful amplification of HIV-1. The A3G-Vif interaction is therefore a key event of viral immune suppression to overcome host defenses, and accordingly, it is an attractive target for new therapeutics against HIV-1/AIDS.

A3G is one of seven APOBEC3 (A3) enzyme family members, A3A, A3B, A3C, A3D, A3F, A3G and A3H, and it is the most potent anti-HIV-1 factor, in the absence of Vif[20,21]. It has duplicated domains, the N-terminal (NTD) and C-terminal domains (CTD), which share a tertiary structure with a preserved zinc-binding motif[22,23]. Structure based mutagenesis analyses have shown that A3G NTD amino acids are

**Table 1 | Polynucleotide sequence of ssDNA or ssRNA used in this study**

| | |
|---|---|
| RNA-I-30 | rGrCrUrArArArCrUrGrArCrGrGrArArGrArUrCrGrUrUrGrGrArArCrArArA |
| RNA-I-25 | rArCrUrGrArCrGrGrArArGrArUrCrGrUrUrGrGrArArCrArArA |
| RNA-I-20 | rCrGrGrArArGrArUrCrGrUrUrGrGrArArCrArArA |
| RNA-I-15 | rGrArUrCrGrUrUrGrGrArArCrArArA |
| RNA-I-10 | rUrUrGrGrArArCrArArA |
| DNA-T25 | dTdTdTdTdTdTdTdTdTdTdTdTdTdTdTdTdTdTdTdTdTdTdTdTdT |
| DNA-I-20 | dCdGdGdAdAdGdAdUdCdGdUdUdGdGdAdAdCdAdAdA |
| RNA-II-20 | rCrGrGrUrUrGrArUrCrGrUrUrGrGrArArCrArArA |
| RNA-II-20a | rCrGrGrArArGrUrUrCrGrUrUrGrGrArArCrArArA |
| RNA-II-20b | rCrGrGrUrUrGrUrCrGrUrUrGrGrArArCrArArA |
| RNA-III-20 (RNA20) | rCrGrGrUrUrGrArUrCrGrUrUrUrArArCrArArA |
| RNA-III-20a | rCrGrGrUrUrGrArUrUrUrUrUrUrArArCrArArA |
| RNA-III-20b | rGrUrUrGrArUrUrUrUrUrUrArArCrArArA |
| RNA-III-20c | rCrGrGrUrUrGrArUrUrUrUrUrArArCrA |
| RNA-III-20d | rCrGrGrUrUrGrArUrUrUrUrUrArArCrArArA |
| RNA-III-20e | rCrGrGrUrUrGrArUrUrGrUrUrU |
| RNA-III-20f | rCrGrGrUrUrGrArUrUrUrUrUrU |
| RNA-IV-20 | rCrGrGrUrUrGrArUrGrUrUrUrUrArArCrArArA |
| DNA-III-20 (DNA20) | dCdGdGdUdUdGdAdUdCdGdUdUdUdUdAdAdCdAdAdA |
| U20 | rUrUrUrUrUrUrUrUrUrUrUrUrUrUrUrUrUrUrUrU |
| U20-rGrA | rUrUrUrUrGrArUrUrUrUrUrUrUrUrUrUrUrUrUrU |
| U20-rUrA | rUrUrUrUrUrArUrUrUrUrUrUrUrUrUrUrUrUrUrU |
| U20-rGrU | rUrUrUrUrGrUrUrUrUrUrUrUrUrUrUrUrUrUrUrU |
| U20-dGrA | rUrUrUrUrUdGrArUrUrUrUrUrUrUrUrUrUrUrUrU |

recognized by HIV-1 Vif[23,24]. Specifically, aspartate-128 (D128) of A3G has been identified as a species-specific determinant since the single mutation, D128K, makes human A3G insensitive to HIV-1 Vif-induced degradation, while Old World monkey A3Gs containing K128 are sensitive to SIV Vif[25–28]. The A3G NTD is also indispensable for incorporation into virions via RNA binding[29–32], whereas the A3G CTD is central to its deamination activity[33,34]. The crystal structure of the A3G CTD in complex with ssDNA has revealed the molecular mechanism of substrate recognition[35]. The active site pocket next to the zinc-binding motif accepts the target cytosine base $dC_{(0)}$, converting it to uracil. Additionally, the preceding base $dC_{(-1)}$ is accommodated in a surface cavity of the A3G CTD to achieve strict sequence specificity; i.e., the dinucleotide $dC_{(-1)}dC_{(0)}$ is essential for A3G recognition. Despite unveiling structure-activity relationships of the CTD, molecular insights into the interaction between A3G NTD, Vif or RNA remain unknown. The difficulty of sample preparation, such as target heterogeneity, has prevented acquisition of high-quality data[36–38].

In this study, we introduce preparation of a solubility enhanced human A3G (sA3G) to overcome target protein heterogeneity (Fig. 1, Supplementary Note 1, and Supplementary Fig. 1). sA3G is sensitive to Vif-induced degradation by wild-type Vif and is not degraded by the Vif (H42A/H43A) variant that is deficient to bind wild-type A3G[39] (Fig. 1e); therefore sA3G represents the wild-type A3G-Vif interaction. Previously, we identified RNA oligomers from the complex containing a solubility enhanced A3G-NTD (A3G-sNTD)[23] and Vif-CBFβ-ELOB-ELOC (VCBC), (Supplementary Fig. 2 and Supplementary Note 2). The RNA oligomer sequence is optimized to generate RNA ligand RNA20 (Table 1) that forms a stable complex with sA3G and a previously optimized Vif-CBFβ (VC) construct[40] (Supplementary Fig. 3a–d). This stable sA3G-VC-RNA20 complex shows a highly homogeneous particle distribution using negative-staining electron microscopy without any

chemical crosslink between the components (Supplementary Fig. 3e). Electron cryo-microscopy (cryo-EM) data of the sA3G-VC-RNA20 complex are collected in a frozen, hydrated state and reconstructed the three-dimensional (3D) electron-potential map by single-particle image processing (Supplementary Figs. 4 and 5 and Supplementary Note 3). The refined C2-symmetrized map reaches 2.8 Å resolution (Supplementary Fig. 6) and allows us to build an atomic model unambiguously (Fig. 1f, g and Supplementary Fig. 7). Our cryo-EM structure shows interactions between A3G-Vif, A3G-RNA and Vif-RNA, and in vitro ubiquitination assays identify key adenine and guanine bases for the Vif-induced ubiquitination of A3G.

## Results

### Structure of the sA3G-VC-RNA interfaces

In the asymmetric unit of this sA3G-VC-RNA20 structure, a single sA3G assembles with two VCs, named VC[red] and VC[blue]. Both VCs contact sA3G via their Vif protein and occupy a significant portion of the sA3G NTD surface area. The ligand, RNA20, participates in both interfaces sA3G-Vif[red] and sA3G-Vif[blue]. Interestingly, two heteromers of sA3G-VC-RNA20 face each other with their sA3G NTD domains (Fig. 1f), forming a dimer contact between two heteromers. This dimer contact is mediated by sA3G NTD residues (75-SKWKLHRD-83) interacting with Vif[blue] residues (76-ERDW-79) and CBFβ[blue] residues (33-RDRP-36) from the other heteromer. The biological relevance of this dimeric arrangement is unclear because preceding residues in wild-type A3G 71-FHWF-74 were all mutated and R78 has been deleted in sA3G (Supplementary Fig. 1e). Since Vif[blue] and CBFβ[blue] are involved in both of intra- and inter-heteromeric interactions, sA3G-VC[blue] formation may be an artifact caused by these mutations and the Vif[blue] contact within each heteromer may be unable to stand on its own without support from the sA3G-Vif[blue] dimer contact. On the other hand, the sA3G-VC[red]-RNA20 interface involves neither mutations nor heterodimer interactions; therefore, it most likely reveals biologically relevant interactions. It is noteworthy that a similar heterodimeric arrangement was found in a certain state, called as state 1, of human wild-type A3G-Vif complex purified from insect cells[41]. This sA3G-VC-RNA20 heterodimer interaction may have stabilized the entire complex and enabled us to obtain high-quality data, which represents the long-sought atomic structure of the A3G-Vif complex, including visualization of side chains at the interfaces (Figs. 1f, g and 2).

The protein-protein interface of sA3G-Vif[red] is biologically pivotal for complex formation because it includes A3G amino acid D128, which has been identified as a determinant of species-specific A3G-Vif interaction[25–28]. Our cryo-EM structure shows sA3G side chains D128 and D130 proximal to the side chain of R15 located in Vif[red] helix α1, suggesting that their rotamers form bifurcated hydrogen bonds (Fig. 2a). This aspartic acid dyad also faces W70/G71 on Vif[red] and is thus essentially surrounded by amino acids from Vif[red]. The preceding sA3G residue, W127, makes contact with Vif[red] H43, forming a π-π interaction between the aromatic rings (Fig. 2a). Vif[red] Y44 makes contact K270, located at the C-terminal end of helix α2 in the CTD (Fig. 2a, c). These interactions support previous findings that Vif residues 40-YRHHY-44 are critical for A3G binding and degradation[39]. Helix α2 of the CTD is arranged along helix α6 of the NTD and interacts mainly via hydrophobic interactions, such as the contact between M188 and F268. Although the sA3G CTD provides a limited interface for the sA3G-Vif[red] assembly, it is likely to have a significant impact on stabilizing the complex, considering that the C-terminal part of CTD helix α2 features the best-defined map resolution in the CTD, i.e., side chain rotamers of F268 and K270 were clearly resolved, as well as their interacting Vif[red] residues and RNA (Fig. 2a, c). The sA3G CTD and the sA3G-Vif[red] assembly support each other, and complex formation defines the CTD arrangement despite the flexible character of both A3G NTD and CTD in solution[36].

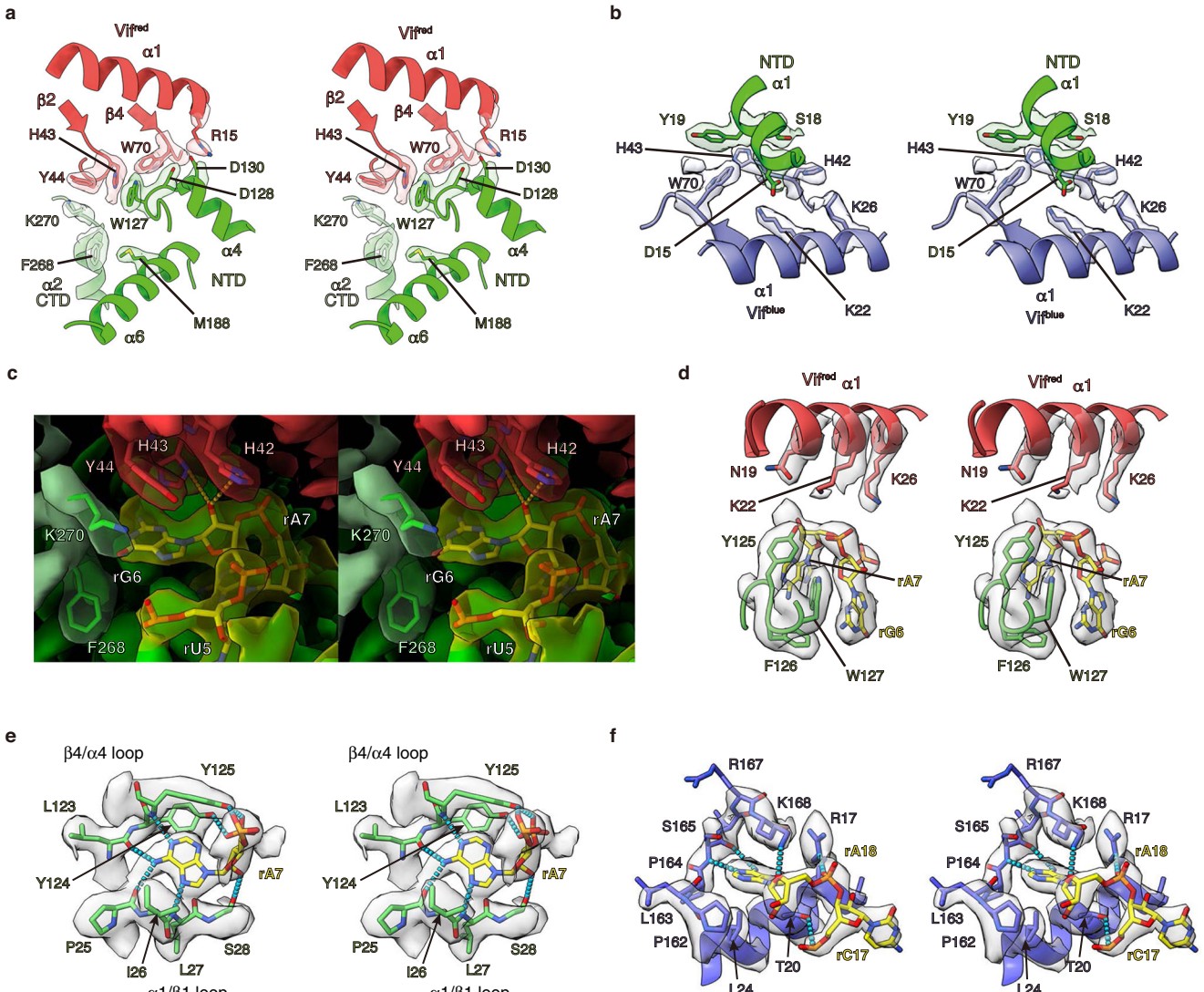

**Fig. 2 | Molecular detail of RNA binding to sA3G and Vif. a, b** Close-up stereo views of the protein-protein interface of sA3G-Vif[red] (a) and sA3G-Vif[blue] (b). Poly-peptides are shown in ribbon representation and selected side chains are depicted with stick models. The cryo-EM map was contoured at a threshold of 5σ and the isoelectron potential surface from these side chains is overlaid. The coloring scheme follows Fig. 1f. **c, d** Close-up stereo views of RNA nucleotides rG6rA7. The cryo-EM map is contoured at 5σ and colored in the same manner as Fig. 1f. Selected side chains and nucleotides are presented as stick models and labeled. Predicted hydrogen bonds between the ribose 2'-hydroxyl group and side chains H42/H43 of Vif[red] are represented by dashed lines (**d**). **e, f** Stereo views of the atomic model showing nucleotide rA7 (**e**) and rA18 (**f**) accommodations in protein surface pockets of sA3G and Vif[blue], respectively. Polypeptides and nucleotides are shown as stick models or in ribbon representation. The cryo-EM map is contoured at 5σ. Predicted hydrogen bonds between protein and nucleotides are depicted by cyan-colored dashed lines. Amino acids P162, L163 and P164 are part of the PPLP motif.

In the protein-protein interface of sA3G-Vif[blue], on the other hand, Vif[blue] interacts with sA3G NTD helix α1 (Fig. 2b). Amino acid D15 of sA3G contacts Vif[blue] K22 on α1. Vif amino acids K22 and K26, speci-fically the positive charge of K26, are required for Vif-induced degra-dation of A3G[42,43]. Additionally, sA3G S18 and Y19 interact with Vif[blue] H43. The side chain of Y19 also faces the aromatic ring of Vif[blue] W70, forming a π-π interaction. Vif mutants W70A and W70R fail to neu-tralize antiviral activity of A3G[44,45]. It is noteworthy that Vif[blue] H43 and W70 contact sA3G Y19, whereas the same Vif residues face sA3G W127 and D128 in the sA3G-Vif[red] interface.

### RNA significance in sA3G-Vif interaction

The areas that account for protein-protein interactions are ~550 Å² and ~590 Å² for sA3G-Vif[red] and sA3G-Vif[blue], respectively. These relatively small interaction surfaces may not be sufficient to form a stable complex. Indeed, no sA3G-VC complex was captured when protein components alone were mixed together, whereas addition of an RNA oligomer induced stable complex formation (Supplementary Fig. 3a–d and Supplementary Note 3). The cryo-EM map explicitly shows that the ligand RNA20 interacts with both interfaces, sA3G-Vif[red] and sA3G-Vif[blue] (Fig. 1f and Supplementary Fig. 9). Specifically, we found two dinucleotides, rG6rA7 and rC17rA18, that were well defined in protein pockets of sA3G NTD and Vif[blue], respectively (Fig. 2c–f).

The dinucleotide rG6rA7 is located at the sA3G-Vif[red] interface (Fig. 2c, d). A binding pocket for rG6 is formed by W127 (NTD), F268, and K270 (CTD) of sA3G and H43 and Y44 of Vif[red]. As described above, these amino acids contact each other to form the protein-protein interfaces between sA3G NTD-Vif[red], sA3G CTD-Vif[red] and sA3G NTD-CTD (Fig. 2a). Additionally, the side chains of Vif[red] K22 and K26 are within hydrogen bonding distance to a phosphate group on the RNA backbone (Fig. 2d). The well-resolved nucleotide, rA7, is buried deeper and is accommodated in the pocket of sA3G NTD formed by NTD loops α1/β1 and β4/α4 (Fig. 2e). The refined atomic model indicates multiple hydrogen bonds between nucleobase of rA7 and backbone atoms of

the loops. In particular, atom N6 of base rA7 is within range to form bifurcated hydrogen bonds with carbonyl oxygens of both P25 and L123. This arrangement suggests a binding preference of the pocket for specific bases. Although the pocket size is large enough to accept a purine base, guanine atom O6 would be repelled by the carbonyl oxygens of P25 and L123, resulting in selective accommodation of adenine base. In addition, the 2′-hydroxyl group of rG6 ribose can form bifurcated hydrogen bonds with side chains H42 and H43 of Vif[red] (Fig. 2c). These intermolecular interactions would compensate for an entropic penalty upon the complex assembly to confer base-specificity of the RNA ligand.

On the other hand, rC17rA18 binds to Vif[blue]. The cryo-EM map shows that the base rA18 is accommodated in a cavity formed by amino acids R17, T20, L24 and P162 to K168 of Vif[blue] including the PPLP motif (Fig. 2f). Previous studies found that mutations of the Vif PPLP motif reduced A3G binding, and increased A3G incorporation to virions[46]. The refined model indicates possible hydrogen bonds between atom N6 of rA18 and the main chain carbonyl groups of Vif[blue] S165 (Fig. 2f). The molecular arrangement of this pocket likely excludes the O6 atom of guanine. Adenine satisfies the geometry and interactions with chemical moieties of surrounding amino acids. In contrast, nucleotide rC17 is exposed to the protein exterior. The atomic model shows that phosphate groups of rC17 and rA18 are within hydrogen bonding distance of side chains of T20 and R17, respectively.

We found that complex formation is also determined by non-specific sA3G-RNA interactions, not only through specific adenine base recognition mentioned above. Although the model assignment to the map was somewhat ambiguous for nucleotides rC1 to rU5, and rU8 to rA16 (Supplementary Fig. 7m), map features for the RNA20 ligand could be rationally interpreted. At a map contour level of 5σ, rC1 to rU5 and rU8 to rA16 were clearly visible at the sA3G-Vif[red] and sA3G-Vif[blue] interfaces, respectively (Supplementary Fig. 9). The cryo-EM map locates base rC1 close to rU8, and the 5′-half of the RNA20, C1 to rU8, adopts a loop conformation (Supplementary Fig. 9e, i, j). Interestingly, nucleotides rC1 and rG2 contact protein surfaces of Vif[red] and even CBFβ[blue] (Supplementary Fig. 9e), apparently filling the gap between sA3G, Vif[red] and CBFβ[blue] and increasing the interface area of the assembly. Nucleotides rU8 to rA16 seem to extend the polynucleotide chain on the surface of sA3G (Supplementary Fig. 9i, j, m, n). Amino acid Y59 of sA3G is not involved in protein-protein interaction, but contacts rU13 and rU14 (Supplementary Fig. 9m, n). The Y59D mutant attenuated Vif-induced degradation, compared with that of wild-type A3G (Supplementary Fig. 1f, lanes 6 and 7). As expected, the introduced aspartic acid most likely excludes accommodation of the RNA due to a repulsive interaction between its negatively charged side chain and phosphates in the polynucleotide backbone. Shirakawa et al. reported that phosphorylation of A3G T32 disrupted Vif-induced degradation[47]. Since T32 is located proximal to Y59 and contacts the ligand RNA (Supplementary Fig. 9n), the impact of T32 phosphorylation on degradation is also likely caused by exclusion of the ligand RNA. Although these amino acids are located away from the sA3G-Vif interface, they mediate complex formation indirectly through interactions with RNA.

Our cryo-EM map and biochemical data demonstrate that RNA mediates both sA3G-Vif[red] and sA3G-Vif[blue] assemblies. The RNA20 ligand increases the interface areas by up to -1500 Å² and -950 Å² for sA3G-Vif[red]-RNA20 and sA3G-Vif[blue]-RNA20, respectively, thus maximizing domain interactions and stabilizing the complex. It explains how RNA20 captures the sA3G-VC complex and enables structure determination by cryo-EM. Intriguingly, only the two nucleotides, rA7 and rA18, participate in base-specific interactions with their protein binding partners. Therefore, sA3G-VC assembly can be promoted by various RNA sequences. We further assessed base specificity and impact on sA3G ubiquitination, as will be discussed hereafter.

## Mechanism of sA3G targeting by Vif

Our cryo-EM reconstruction of the sA3G-VC-RNA20 complex reveals details of interaction between Vif, RNA and sA3G. To visualize those interactions at a glance, all intermolecular contacts are summarized in Fig. 3a. Amino acid determinants of sA3G form patches that can be identified as specific interfaces for either Vif[red], Vif[blue] or RNA (Fig. 3a, b, d); e.g., residue D128 of sA3G participates exclusively in the interface with Vif[red] (Fig. 3a, b). In contrast, Vif[red] and Vif[blue] share amino acid determinants that can bind to different portions of sA3G or RNA, i.e., amino acid R15 of Vif[red] contacts D128, D130 and Y131 of sA3G and the same residue of Vif[blue] makes contacts with rU11 and rU14 (Fig. 3a, c, e). Amino acids R15, W79, L81 and Q83 of Vif[red] interact with sA3G, while those of Vif[blue] interact with RNA. Conversely, amino acids R23, K26, Y30 and H42 of Vif[red] participate in the interface with the ligand RNA while those of Vif[blue] are involved in sA3G interaction. Thus, these amino acid determinants of Vif form a complementary target pattern on each interface; i.e., they differ between the interfaces of sA3G-Vif[red] and sA3G-Vif[blue], although both Vif[red] and Vif[blue] use the same face of their protein surface to the interface. Only Vif H43 and W70 interact with sA3G in both interfaces (Figs. 2a, b and 3a), and these two amino acids have been identified as critical determinants for A3G-Vif interaction[39,45].

To understand the influence of electrostatic potential effects in the A3G recognition by Vif, we calculated the electrostatic surface charge distribution on sA3G and both Vifs (Fig. 3f–i). The electrostatic surface potential distribution shows that sA3G is dominated by negative potentials on both Vif binding patches (Fig. 3f), whereas the RNA binding site stands out as a positively charged area (Fig. 3f), supporting that RNA binds sA3G mainly by electrostatic interactions. Upon RNA binding, the electrostatic surface potential of the sA3G-RNA20 complex becomes an extended region of predominantly negative potentials (Fig. 3g). This enhancement of negative potentials will promote an association with positive potentials on the interaction surfaces of Vif (Fig. 3h, i). Thus, RNA mediates electrostatic complementarity between sA3G and Vif, and this electrostatic complementarity is most likely the major factor driving assembly of sA3G and Vif. In addition, a predicted model of wild-type A3G suggests that the wild-type amino acids R14, E170 and E173 increase the electrostatic potentials and enhance the electrostatic complementary between A3G, Vif and RNA (Supplementary Fig. 8a, f, g).

## RNA promotes sA3G ubiquitination

How does RNA base specificity affect Vif-induced ubiquitination of A3G? To test this, we explored in vitro ubiquitination using the ubiquitin-activating enzymes E1, ubiquitin-conjugating enzyme E2 L3, NEDDylated CUL5, and VCBC proteins. In the presence of all required proteins, sA3G was polyubiquitinated, as expected, showing ladder-like bands on PAGE gels (Fig. 4a, lanes 8 and 9), whereas no ubiquitinated sA3G was detected when a protein component was missing (either substrate sA3G or ARIH2), or when using ubiquitin lacking glycine-76 at the C-terminus and is incapable to form an isopeptide bond with the substrate protein (Fig. 4a, lanes 1–4). Although the reaction occurred in the absence of the RNA20 ligand (Fig. 4a, lane 5), it was much less efficient than when RNA20 was present (Fig. 4a, lane 8). In addition, we tested A3G amino acid replacements which were previously identified to reduce Vif-induced degradation, i.e., mutations D128K or K297R/K301R/K303R/K334R[25–27,48]. These mutations appeared to decrease polyubiquitination (Fig. 4a, lanes 6 and 7).

We further assessed effects of the RNA20 ligand and sA3G mutation on the ubiquitination reaction by monitoring the amount of intact sA3G remaining. Under our test conditions, sA3G was almost completely ubiquitinated within the monitoring period (Fig. 4b lanes 1–4, and 4c). As expected, mutation D128K slowed mono- and di-ubiquitination of sA3G (Fig. 4b lanes 5–8, and 4c). Interestingly, a DNA oligomer with a sequence corresponding to RNA20 showed no ability to promote sA3G ubiquitination (Fig. 4b lanes 9–12, and 4c). These results clearly indicate that RNA20 is the driver of sA3G ubiquitination.

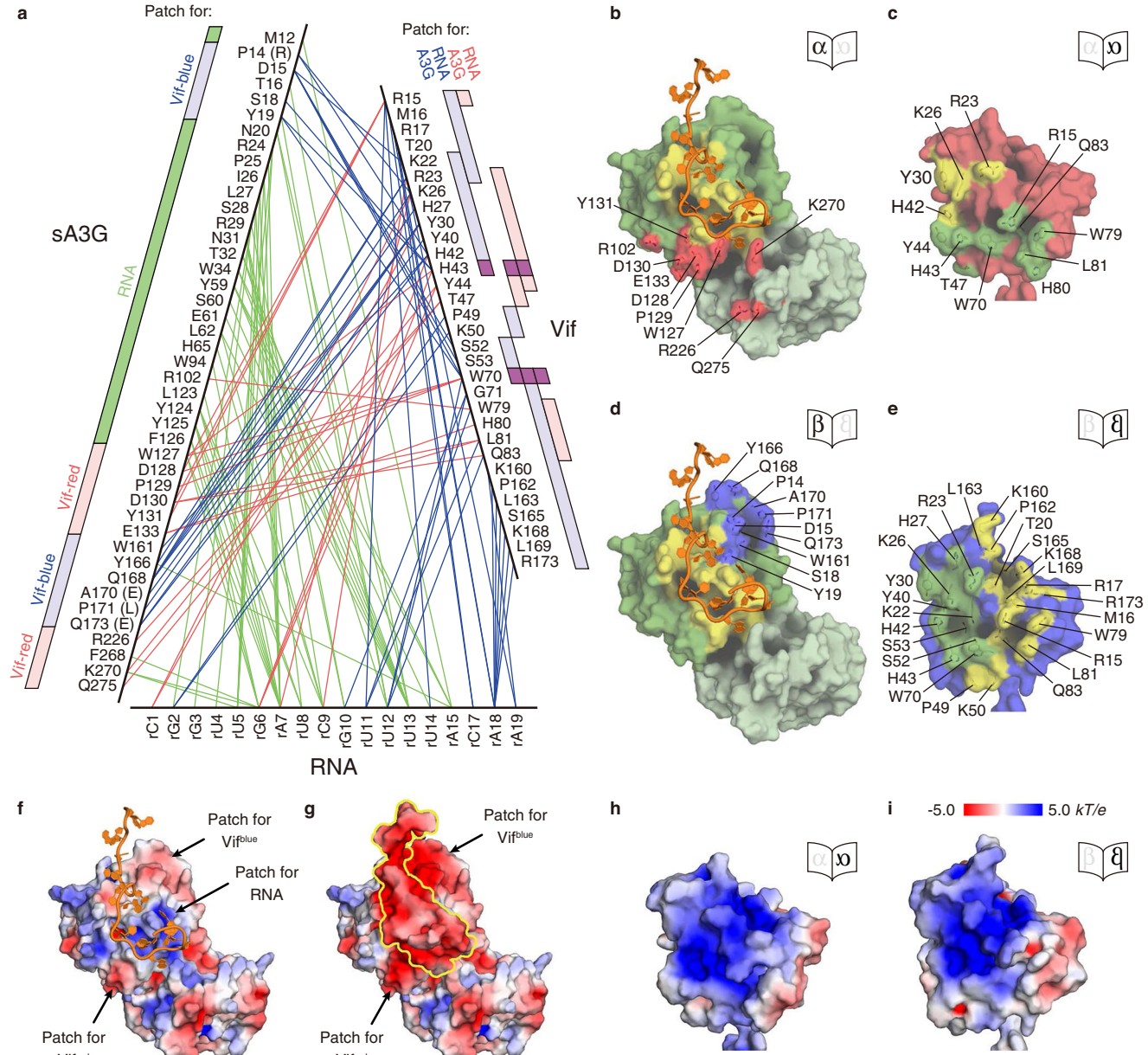

**Fig. 3 | Molecular interface features promoting sA3G-VC-RNA20 complex formation. a** Overview diagram of interactions between complex components, sA3G, Vif and RNA20. Inter-residue contacts are indicated by lines: sA3G-RNA20 (green), sA3G-Vif$^{red}$ or RNA20-Vif$^{red}$ (red), and sA3G-Vif$^{blue}$ or RNA20-Vif$^{blue}$ (blue). Wild-type A3G amino acids are denoted in parentheses. Binding sites on sA3G with Vif$^{red}$, Vif$^{blue}$ and RNA20 are indicated by rectangles colored light red, light blue and green, respectively (left). Binding interfaces on Vif$^{red}$ and Vif$^{blue}$ are represented by rectangles colored light red and light blue, respectively (right). Vif amino acids H43 and W70 are highlighted in purple. **b–e** Solvent-accessible surface presentations of sA3G (**b**, **d**), Vif$^{red}$ (**c**) and Vif$^{blue}$ (**e**) showing binding interfaces. Coloring of sA3G and Vifs follows Fig. 1f. The RNA20 model is drawn in cartoon representation, colored orange. In the sA3G-Vif$^{red}$ interface, binding sites on sA3G with Vif$^{red}$ and RNA20 are colored red and yellow, respectively (**b**), whereas interaction sites on Vif$^{red}$ with sA3G and RNA20 are colored green and yellow, respectively (**c**). In the sA3G-Vif$^{blue}$ interface, sA3G interacts with Vif$^{blue}$ and RNA20 via regions colored blue and yellow, respectively (**d**), while Vif$^{blue}$ binds to sA3G and RNA20 through regions colored green and yellow, respectively (**e**). **f–i** Electrostatic potential surface representations of sA3G without (**f**) and with (**g**) RNA20, Vif$^{red}$ (**h**) and Vif$^{blue}$ (**i**). Calculated surface potentials are colored with a gradient from red (negative) to blue (positive). Binding interface patches are labeled (**f**, **g**). The position of the ligand, RNA20, is outlined in yellow (**g**). The electrostatic potential scale is indicated in (**i**).

As mentioned above, a dinucleotide rG6rA7 of the ligand RNA20 forms multiple hydrogen bonds between sA3G and Vif$^{red}$ (Fig. 2c, e). The rG6rA7 interaction is important since sA3G ubiquitination remains enhanced when using an RNA oligomer with a poly-uridine sequence, except for rG6rA7 (U20-rGrA) although the reaction proceeds slowly (Fig. 4d lanes 1–4, and 4e, Table 1). Whereas rG6-to-rU6 replacement was still able to promote the reaction (Fig. 4d lanes 5–8, and 4e), omission of nucleotide rA7 no longer enhanced the reaction significantly (Fig. 4e). Intriguingly, a pinpoint modification, U20-dGrA,

i.e., removal of the 2′-hydroxyl group of rG6, led to a significant loss of enhancement (Fig. 4d, lanes 9–12 and 4e). As described above, the rA7 base is accommodated in a deep pocket of sA3G and the 2′-hydroxyl group of rG6 can form bifurcated hydrogen bonds with the side chains of H42 and H43 on Vif$^{red}$, so that a lack of either one severely weakens the interaction with sA3G-Vif$^{red}$ (Fig. 4e, f). The importance of these histidines was confirmed by introducing H42A/H43A mutations on Vif, which abolished degradation of wild-type A3G and sA3G in cells (Supplementary Fig. 1f, lanes 5 and 15)[39].

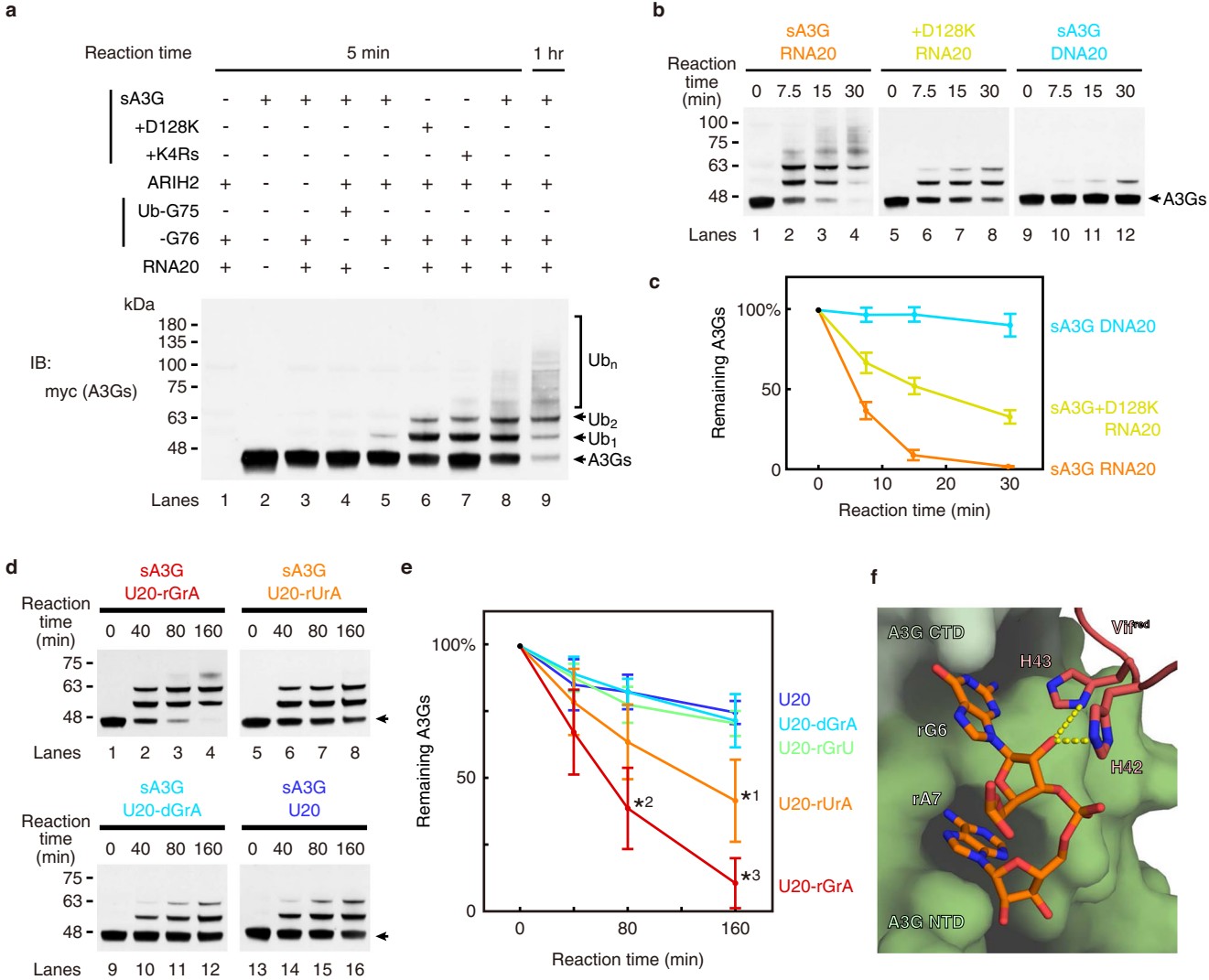

**Fig. 4 | In vitro ubiquitination of sA3G and effects of polynucleotide ligands.**
**a** In vitro ubiquitination assay using sA3G and its mutants. Substrates were specifically detected using a C-terminally-tagged c-myc sequence (lanes 1 and 2). Ubiquitination of sA3G required ARIH2 and ubiquitin with a C-terminal G76 (lanes 3 and 4). The reaction was enhanced in the presence of RNA20 (compare lanes 5 and 8). Amino acid mutations of sA3G, D128K (lane 6) and K4Rs (K297R/K301R/K303R/K334R) (lane 7) appeared to reduce polyubiquitination beyond diubiquitin. Most of sA3G was ubiquitinated within 1 h of reaction (lane 9). **b** Time course of sA3G ubiquitination. The band of unreacted sA3G is indicated by an arrow. In the presence of RNA20, sA3G was depleted within 30 min (lanes 1–4), whereas the ubiquitination was attenuated by sA3G mutation D128K (lanes 5–8). DNA20 showed almost no sA3G ubiquitination (lanes 9–12). **c** Quantification of unreacted sA3G during the reaction presented in (**b**). Color coding follows (**b**). Assays were performed independently in triplicate ($n = 3$). Data points represent mean values. Error bars indicate standard deviation. **d, e** Impact of dinucleotide rG6rA7 on sA3G ubiquitination. Reactants were analyzed by western blotting (**d**). The band of

unreacted sA3G is indicated by an arrow. The amount of unreacted sA3G was quantified (**e**). A significant ($p < 0.05$ by two-sided $t$ test, no adjustments) decrease in unreacted sA3G is indicated by an asterisk compared with that in presence of U20; $p$ values are 0.0095, 0.0024 and 0.000012 for *1, *2 and *3, respectively. During monitoring, sA3G was depleted in the presence of U20-rGrA (**d**, lanes 1–4), whereas U20 showed a much slower reaction (**d**, lanes 13–16). U20-rUrA retained some ability to enhance sA3G ubiquitination (**d**, lanes 5–8). In contrast, U20-dGrA (**d**, lanes 9–12) and U20-rGrU (**e**) significantly lost the ability to enhance sA3G ubiquitination. Assays were performed independently in triplicate ($n = 3$). Data points represent mean values. Error bars indicate standard deviation. **f** Close-up view of dinucleotide rG6rA7 at the sA3G-Vif[red] interface. The solvent-accessible surface of sA3G is shown in green. The dinucleotide, rG6rA7, and side chains of Vif[red] H42/H43 are drawn in stick representation and labeled. Predicted hydrogen bonds are depicted as dashed yellow lines. Source data are provided as a Source data file (**a**–**e**).

We revealed two well-resolved dinucleotides rG6rA7 and rC17rA18 in the cryo-EM map. Their characteristic molecular envelopes (Fig. 2c–f) and unique occurrence among systematically tested RNA ligands (Supplementary Fig. 3a–d) allowed unambiguous assignment. During RNA sequence optimization to capture a stable sA3G-VC complex, we found that replacement of rA7-to-rU7 caused complete loss of forming the sA3G-VC complex (Supplementary Fig. 3b), while a partial truncation of ligand RNA, including removal of rC17rA18, still allowed to form the sA3G-VC complex (Supplementary Fig. 3d). rG10 was partially able to compensate for the loss of rC17rA18 (Supplementary Fig. 3d).

Interestingly, a systematic truncation of 3′-side indicates that a certain length of polynucleotide is required for enhancing the ubiquitination rather than the specific nucleotides rC17rA18. This is because U13-rGrA and U11-rGrA which are lacking rC17rA18 can promote the ubiquitination as well as U20-rGrA, yet further truncations, U9-rGrA and U7-rGrA, no longer support the A3G ubiquitination effectively (Supplementary Fig. 10). The molecular models show that U9-rGrA and U7-rGrA are too short to provide a large negatively charged surface for Vif interaction.

Taken together, our in vitro ubiquitination and in cell degradation assays showed that the dinucleotide rG6rA7 serves an essential

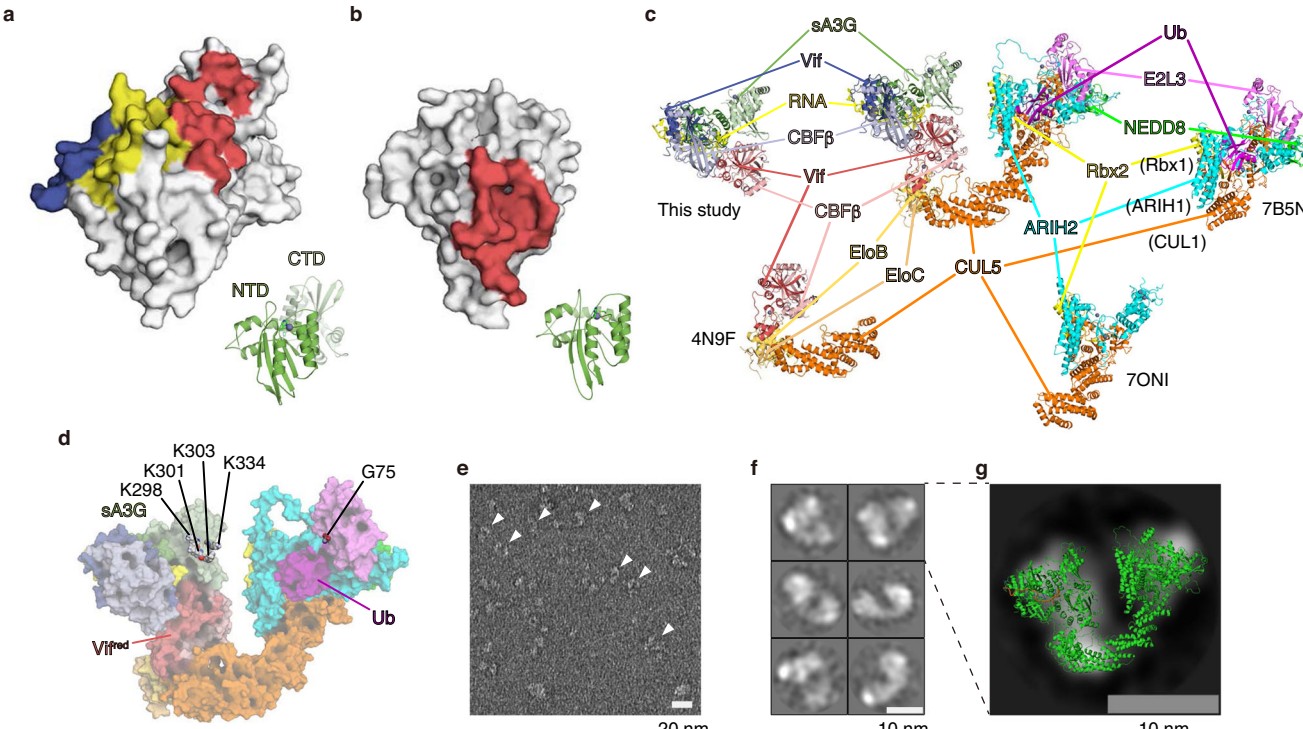

**Fig. 5 | Model of A3G captured in a ubiquitin ligase complex. a, b** Comparison of Vif-binding interfaces from sA3G (**a**) (this study) and A3F (**b**)[40]. The molecular orientation of A3 proteins is indicated using a cartoon representation (bottom right inset). Binding interfaces on sA3G with Vif[red], Vif[blue] and RNA are shown in red, blue and yellow, respectively (**a**), whereas the interface on A3F with a single Vif is colored red (**b**). **c** Predicted model of the entire ubiquitin ligase complex. The model was built using alignments of atomic coordinates obtained from our cryo-EM structure and PDB IDs 4N9F, 7B5N and 7ONI. Each component is labeled. **d** Solvent-accessible surface representations of the predicted ubiquitin ligase complex linked through sA3G-Vif[red]-CUL5. Amino acids sA3G K297/K301/K303/K334 and ubiquitin G75 are drawn as space-filled models and labeled. **e** Unprocessed TEM micrograph of a negatively stained sample from a mixture of in vitro ubiquitination assay reactant. The image reflects many states of the ongoing reaction. Arrowheads indicate possible U-shaped ubiquitin ligase complex particles. Scalebar, 20 nm. **f** 2D class average images of putative U-shaped ubiquitin ligase complex particles. The 6 classes were composed of 1095 particles harvested from 179 micrographs. Scalebar, 10 nm. **g** Superimposition of the predicted ubiquitin ligase complex model (**d**) on a class average image from (**f**). Scalebar, 10 nm.

biological function in conferring local base preference upon the RNA ligand to promote A3G-Vif interaction. The length of RNA ligand is likely important for the enhancement of the A3G-Vif interaction.

## Discussion

We determined binding a mode of ssRNA for sA3G at the interface of sA3G-Vif. Is this ssRNA binding-mode a common feature in A3 proteins? A3 proteins commonly target cytosine on ssDNA for conversion into uracil. The hotspot sequence has been clearly defined: $dC_{(-1)}dC_{(0)}$ for A3G and $dT_{(-1)}dC_{(0)}$ for other A3 enzymes. The target cytosine base $dC_{(0)}$ is inserted into an active-site pocket proximal to a zinc-binding motif, whereas the $dC_{(-1)}/dT_{(-1)}$ is accommodated in a cavity formed by loops corresponding to α1/β1 and β4/α4 in the A3G CTD, which determines target sequence specificity (Supplementary Fig. 11c, d)[35,49]. Additionally, some A3 structures were reported in complex with ssDNA or ssRNA, although these oligonucleotides were not placed in the catalytic pocket (Supplementary Fig. 11a, b, e)[50–52]. Interestingly, upon binding in an enzymatically active or inactive mode, a single base of the polynucleotide ligand is recognized by the cavity formed between loops α1/β1 and β4/α4. Our cryo-EM structures consistently showed that base rA7 of ssRNA is accommodated in a cavity formed by corresponding loops of sA3G NTD (Supplementary Fig. 11f). The structural similarities explicitly indicate that the observed ssRNA binding mode to sA3G can be attributed to an essential molecular feature of A3 enzymes because the 5′ to 3′ directionality of the RNA ligand bound to sA3G is shared by substrate binding of enzymatically active A3s, A3G shown by CTD-ssDNA and A3A-ssDNA co-crystal structures (Supplementary Fig. 11c, d, f). We tested deaminase activity

on ssRNA by sA3G, but no rC deaminase reaction was observed using in vitro deamination assay, even though base rC9 points toward the pseudo-catalytic pocket of sA3G NTD (Supplementary Fig. 9i).

RNA binding to A3G has been discussed since the early stages of A3G research[10]. After expression, A3G binds to cellular or viral RNAs, and it forms high-molecular-mass (HMM) protein-RNA complexes in cells. The RNA sequence preference of A3G has been thoroughly studied in the context of A3G-Gag interaction rather than A3G-Vif assembly. However, no consensus RNA sequence was found, although a weak preference for an rA/rG-rich sequence has been suggested[53,54], and Apolonia et al. concluded that A3G binds to RNAs promiscuously[55]. In our study, we found that rG6rA7 is a critical determinant of the sA3G-VC assembly. Since various cellular and viral RNAs contain rG/rA rich sequences[53–55], it is plausible that A3G-Vif and A3G-Gag interactions may compete with each other and reduce encapsidation of A3G.

Our sA3G-VC-RNA20 structure indicates that the sA3G-Vif interface is consistent with the previously identified wild-type human A3G-Vif interface (Fig. 5a)[24] while it differs from that of A3F CTD-Vif (Fig. 5b)[40]. Ito et al. have recently presented a cryo-EM model of Rhesus macaque A3G (rmA3G) with CRL5[Vif-CBFβ] E3 ubiquitin ligase complex at 3.1 Å resolution[56]. Their cryo-EM structure revealed that double-stranded RNA with a flanking single-stranded region at the 3′-end can form a rmA3G-dsRNA-Vif complex. The rmA3G-dsRNA-Vif structure showed that dinucleotide R+1 and R+2 in the single-stranded region of the RNA mediates the A3G-Vif interaction in a similar manner by which rG6rA7 binds sA3G and Vif[red][56]. Most recently, a cryo-EM structure of human A3G (hA3G) complexed with Vif-CBFβ-EloB-EloC (VCBC) was determined by Li et al[41]. This hA3G-VCBC structure also revealed single-stranded RNA

bound at the A3G-Vif interface, and hA3G-RNA-Vif interactions were very similar to that found in our sA3G-RNA20-Vif[red] structure. Therefore, three independent structures containing different A3Gs have revealed almost identical A3G-RNA, Vif-RNA and A3G-Vif interactions, high-lighting the importance of sA3G-RNA20-Vif[red] interaction.

We further assessed our cryo-EM structure for forming a ubiquitin ligase holo-enzyme. Several structures of the ubiquitin ligase complex have been reported[19,57,58], but none of them included the linkage between A3G and Vif with a high-resolution reconstruction. Our structure provides this missing piece to reconstitute a complete holoenzyme. We aligned each component to assemble an atomic model of the complete ubiquitin ligase complex, including E2, E3, VCBC and substrate sA3G (Fig. 5c). Considering that sA3G-Vif[red] inter-action seems to be essential for complex formation and A3G ubiqui-tination, we built a holoenzyme model using the sA3G-RNA20-Vif[red] structure. In this model, A3G lysine residues K297, K301, K303 and K334 which are frequently ubiquitinated[48], face the activated ubiquitin bound to E2 and ARIH2 (Fig. 5d). This arrangement seems to allow for these lysines to access the C-terminus of ubiquitin to form a chemical conjugation between them. Furthermore, this assembly model is cor-roborated by the fact that we frequently found U-shaped molecular envelopes in negative stained EM images of the in vitro ubiquitination reaction mixture that consists of charged E2 ubiquitin ligase and sA3G-RNA20-VCBC-CUL5 E3 ligase complex (Fig. 5e–g). This U-shaped molecular envelope was also found in the recent cryo-EM study of the CRL5[Vif-CBFβ]-ARIH2-A3G complex although the structures of A3G and Vif were not defined due to disorder[58].

In summary, our high-resolution cryo-EM structure of sA3G-VC-RNA20 revealed that Vif binds sA3G through interactions with ssRNA. Our in vitro and in cellulo experiments showed that the sA3G-Vif-RNA trimolecular interface involving the dinucleotide rG6rA7 is most cri-tical for ubiquitination of sA3G; therefore, this trimolecular interface is likely used to form a ubiquitin E3-ligase complex. The good fit of the predicted ubiquitin ligase complex model with 2D class average ima-ges holds promise for determining the structure of the fully assembled ubiquitin ligase complex, which may provide further understanding of structure-activity relationships of A3G ubiquitination.

## Methods

### Molecular cloning and protein purification of sNTD, sA3G, VCBC, VC, ubiquitin ligase complex, and their mutants

The solubilized human A3G NTD construct (sNTD) and a complex Vif-CBFβ-ELOB-ELOC (VCBC), were prepared as described[23]. The Vif-CBFβ (VC) construct was prepared as described[39] with a minor modification (Fig. 1b). Preparation procedures were partially modified and are briefly described as follows.

Expression plasmids for glutathione S-transferase (GST)-fused sNTD or sA3G and its mutants were constructed using pCold-GST vec-tors (a gift from Dr. Kojima, currently distributed by Takara Bio, Japan). E. coli strain BL21 (DE3) was transformed using the plasmid and culti-vated in LB media at 37 °C. At a turbidity of 0.8–1.0 OD, expression was induced at 20 °C by adding 0.2 mM isopropyl-β-thiogalactopyranoside (IPTG). After overnight cultivation, cells were harvested and resus-pended in buffer-1 [50 mM sodium phosphate, pH 7.3, 200 mM sodium chloride, 0.01% Tween 20-, and 0.5-mM Tris(2-carboxyethyl)phosphine hydrochloride (TCEP)]. Cells were lysed by sonication, and the lysate was centrifuged at 20,000 × g for 30 min. The supernatant was applied to glutathione-immobilized resin (Genscript) equilibrated in buffer-1. The protein-bound resin was washed with buffer-2 (50 mM Tris-HCl, pH 8.0, 200 mM sodium chloride, 0.5 mM TCEP). An aliquot of HRV 3 C protease solution (Fujifilm Wako Pure Chemical, Japan) was added to the resin resuspension, and the mixture was incubated at 4 °C over-night. The supernatant was applied to a Superdex 200 column (Cytiva) equilibrated with buffer-3 (25 mM Tris-HCl, pH 8.0, 150 mM sodium chloride, and 0.5 mM TCEP). Fractionated protein was concentrated and

kept in a freezer at −80 °C until further use. Protein purity was verified on 12% or 15% polyacrylamide gels followed by Coomassie brilliant blue staining, or on 4–12% gradient Bis-Tris gels (Invitrogen) stained with SimplyBlue SafeStain (Invitrogen). sA3G mutants and C-terminally c-myc-tagged constructs were prepared in the same manner.

Vif and CBFβ constructs shown in Fig. 1b were inserted into BamHI/HindIII and NdeI/XhoI sites, respectively, on pRSFDuet vectors (Novagen). E. coli strain BL21 (DE3) was transformed using this plasmid and cultivated in LB media at 37 °C. At a turbidity of 0.8 ~ 1.0 OD, 1 mM IPTG was added. Four hours later, cells were harvested and resus-pended in buffer-2 supplemented with 20 mM imidazole and 10% dimethyl sulfoxide (DMSO). Cells were lysed by sonication, and the lysate was centrifuged at 20,000 × g for 30 min. The supernatant was applied to buffer-2-equilibrated Ni-NTA resin (Fujifilm Wako Pure Chemical). After washing the resin, bound proteins were eluted with buffer-2 supplemented with 200 mM imidazole and 500 mM dime-thylethylammonium propane sulfonate, NDSB-195 (Hampton Research). Eluent was loaded onto a Superdex 200 column equili-brated in buffer 2, supplemented with 5% DMSO. Immediately after collecting VC fractions, 500 mM NDSB-195 was added, and the mixture was concentrated and kept in a freezer at −80 °C until further use.

Genes of human cullin-5, CUL5, (amino acid residues S12 to A780), RING-box protein 2, Rbx2, (M1 to K113), ubiquitin-conjugating enzyme E2L3 (M1 to D154), NEDD8 (M1 to G76), ubiquitin (M1 to G76), NEDD8-activating enzyme E1 complex, NAE1 (M1 to L534) and UBA3 (M1 to S463), NEDD8-conjugating enzyme UBE2F (M1 to R185) were codon-optimized for bacterial expression and synthesized (Genscript). Their mutants were constructed using a PCR-based technique. Synthesized DNAs for E2L3, NEDD8, ubiquitin, and UBE2F were individually inser-ted into NdeI/XhoI sites on pCold-GST vectors. Protein expression and purification followed the procedure for sA3G preparation described above. DNAs encoding CUL5 and Rbx2 constructs were inserted into the BamHI/XhoI site on a pGEX-6P-1 vector (Cytiva), and into BamHI/HindIII sites on a pRSFDuet vector (Novagen), respectively. Proteins CUL5 and Rbx2 were co-expressed and co-purified in a manner similar to sA3G preparation. DNAs encoding NAE1 and UBA3 were inserted into BamHI/HindIII and NdeI/XhoI sites, respectively, on pACYCDuet vectors (Novagen). NAE1/UBA3 preparation followed the method for VC purification described above, except for addition of DMSO and NDSB-195. Obtained protein solutions were concentrated and then frozen at −80 °C until further use. We prepared all proteins, their mutants, and truncated variants used in this study. Residue numbering follows wild-type constructs.

### RNA cloning

The fraction of sNTD-F126 in complex with VCBC was prepared for RNA extraction with TRIzol and precipitated with ethanol. The pellet was used for a serial enzymatic reaction and for the cloning: 3′-end of RNA was extended by poly(A) polymerase (New England BioLabs), 5′-adapter (5′-rGrUrUrCrArGrArGrUrUrCrUrArCrArGrUrCrCrGrArCrGrArUrC-3′) was ligated with T4 RNA ligase (New England BioLabs), fol-lowed by reverse transcription with a poly-thymidine DNA oligo, and amplification of the cDNA by polymerase chain reaction (PCR) using Taq DNA polymerase (Takara Bio, Japan). Resulting DNAs were inser-ted into pMD20-T vectors (Takara Bio, Japan), and clones were sub-jected to DNA sequencing.

### Pull-down assay using sNTD variants and VCBC

Purified GST-fused sNTD variants and VCBC were mixed at a final concentration of 10 μM each in phosphate buffer, 25 mM sodium phosphate, pH 7.3, 150 mM sodium chloride, and 0.5 mM TCEP. RNA or DNA oligomer were added at concentration of 20 μM, and the mixture was incubated for 10 min at room temperature. A 20-μL aliquot of 50% glutathione-immobilized resin slurry was added to 100 μL of the mixture and incubated further. The protein bound resin was washed

twice with buffer and harvested. The resin was mixed with 20 µL of PAGE loading buffer (Invitrogen), and the mixture was separated on a 4–12% gradient Bis-Tris gel (Invitrogen) and stained with SimplyBlue safe stain reagent (Invitrogen). The gel was scanned, and band densities were analyzed using NIH ImageJ software[59].

### Preparation of sA3G-VC-RNA complex

Purified sA3G, VC and a synthesized RNA oligomer (see Table 1; IDT technologies, Inc. or Fasmac, Japan) were mixed at a ratio of 1:1.5:1.5 in buffer-3 supplemented with 200 mM NDSB-195. The mixture was applied to a Superdex 200 column equilibrated in buffer-3 and purified by size-exclusion chromatography. The largest fraction of the complex was used for negative staining EM and cryo-EM grid preparations immediately after collection. The relative amount of harvested complex was monitored by UV absorbance at 280 nm ($A_{280}$), and RNA binding was roughly estimated by the $A_{260}/A_{280}$ ratio. The prepared sA3G-VC-RNA complex typically had a ratio of 0.98–1.02.

### In vitro ubiquitination assay

Purified CUL5/Rbx2, NAE1/UBA3, UBE2F, and VCBC were mixed at a final concentration of 5 µM each, with 10 µM NEDD8 and 1 mM of freshly prepared ATP in an assay buffer, 25 mM Tris-HCl, pH 8.0, 150 mM sodium chloride, and 0.01% Tween 20. The mixture was incubated at 20 °C for at least 1 h. NEDDylation of CUL5 was verified by PAGE analysis. A 10-µL aliquot of the resulting mixture was mixed with 1 µM sA3G or its mutant, 5 µM E2L3, 30 µM ubiquitin or its mutant, 20 µM DNA or RNA oligomer, and an additional 1 mM ATP, and the total volume was adjusted to 50 µL with assay buffer. Finally, 0.2 µM UBE1 (Sigma Aldrich) were added. The mixture was incubated at 20 °C, and an aliquot of the reactant was harvested for further analysis after 0, 7.5, 15, and 30 min.

To detect sA3G, its mutants and their ubiquitinated variants selectively, a c-myc-tag sequence was attached at their C-termini and detected by western blotting with primary (monoclonal anti-c-myc tag, 9E11, Abcam) and secondary antibodies (monoclonal anti-mouse IgG, ab216772, conjugated with a fluorophore IRDye 800CW, Abcam). Protein bands were detected with a ChemiDoc system (Bio-Rad Laboratories) and analyzed using NIH ImageJ[59].

### Vif-induced A3G degradation assay in human cells

We followed a method in ref. 23. Briefly, human embryonic kidney (HEK293T) cells in 12-well plates were co-transfected with pcDNA3.1-based A3G, sA3G, or their mutant expression vector (0.66 µg) and pcDNA-HVif, pcDNA-HVif-H41A/H42A or empty pcDNA3.1 vector (1.32 µg) with the transfection reagent, FuGene HD (Promega). After incubation for 24 h, 2 µM MG132 or DMSO was added to the medium, and cells were incubated for an additional 24 h. Resulting cell lysates were analyzed by SDS-PAGE and immunoprobed with monoclonal anti-c-myc (9E11, Abcam), anti-Vif (319, Abcam), or anti-α-tubulin primary antibodies (DM1A, Abcam), and monoclonal anti-mouse IgG secondary antibody conjugated with a fluorophore (ab216772, Abcam). Protein bands were detected with a ChemiDoc system (Bio-Rad Laboratories) and analyzed using NIH ImageJ[59].

### Negative-staining electron microscopy

A 3-µL aliquot of freshly prepared sample was dropped on a carbon-coated, glow-discharged copper grid. After incubation for 1 min, excess liquid was removed by blotting. After washing the grid with 20 mM Tris-HCl (pH 8.0) solution, the attached protein was stained with 2% uranyl acetate solution. The stained sample was examined on a JEM-1230R transmission electron microscope (TEM) operating at 100 kV acceleration voltage (Jeol, Japan) or a Talos L120C TEM operating at 120 kV acceleration voltage (Thermo Fisher Scientific). Images were recorded using a Ceta 2 camera (Thermo Fisher Scientific) with EPU software (Thermo Fisher Scientific). Images were analyzed and processed into 2D class averages with RELION software[60].

### Grid preparation for cryo-EM

The concentration of freshly prepared sA3G-VC-RNA complex solution was adjusted so that the UV absorbance at 280 nm was in the range of 0.30–0.33. A holey EM grid (Quantifoil R1.2/1.3) was plasma-cleaned (Solarus II, Gatan Inc.) and treated with graphene oxide film flakes (Sigma Aldrich). Immediately after sample preparation, 3 µL of sample solution was deposited onto the grid at 4 °C and 100% humidity. The sample grid was blotted with filter paper for 3 s and vitrified by plunging it into ethane/propane at liquid nitrogen temperature on a Vitrobot Mark IV (Thermo Fisher Scientific). Vitrified sample grids were stored in liquid nitrogen until further use.

### Cryo-EM data collection

Cryo-EM movie datasets were collected on a Talos Arctica cryo-TEM (Thermo Fisher Scientific) operating at an acceleration voltage of 200 kV with a Falcon 3EC direct electron detector (Thermo Fisher Scientific) in counting mode for sA3G-VC-RNA-I-20 and sA3G-VC-RNA-II-20; on a Titan Krios G1 cryo-TEM (Thermo Fisher Scientific) operating at an acceleration voltage of 300 kV with Falcon 3EC direct electron detector (Thermo Fisher Scientific) in counting mode for sA3G-VC-RNA20. Movie data for sA3G-VC-RNA-IV-20 were recorded with a Falcon 4 direct electron counting detector (Thermo Fisher Scientific) in electron-event representation (EER) format on a Titan Krios G4 TEM (Thermo Fisher Scientific) operating at 300 kV acceleration voltage, and equipped with a Selectris X energy filter (slit width 10 eV) (Thermo Fisher Scientific). Detailed data collection conditions and statistics are provided in Table 2. All cryo-EM data were collected semi-automatically using EPU software (Thermo Fisher Scientific).

### Cryo-EM image processing and analysis

Collected movie data were imported and subjected to sequential procedures, motion correction and phase contrast transfer function (CTF) estimation using MotionCor2[61] and CTFFIND4[62] programs, respectively. All single-particle data were processed with RELION software[60]. Details of image data processing are provided in the Supplementary Notes 3 and 4 and Supplementary Figs. 4 and 5. For the analysis of sA3G-VC-RNA-I-20, particles were initially picked with Laplacian of Gaussian (LoG) function from a set of 3,160 micrographs, 2D classified to remove junk to obtain an initial clean particle stack (508,997 particles). A subset of 105,608 particles belonging to 16 class sums with fine features (panel in Supplementary Fig. 4) was used to create four ab initio 3D reconstructions. The dominant reconstruction was then used as a reference map for 3D classification of the entire clean stack into eight classes, the best of which reached 4.9 Å resolution after refinement. The non-symmetrized reconstructions clearly contained 2-fold symmetry. This map lowpass-filtered to 20 Å was used for template-based particle picking (the final outcome was not affected, whether lowpass-filtering this template to 20 Å or 60 Å). The full dataset of 9,242 micrographs yielded 780,545 particles after junk removal by 2D classification (Supplementary Fig. 4), which were subjected to 3D classification. The largest 3D class (representing 40% of input particles) was refined with C2 symmetry. The map resolution of sA3G-VC-RNA-I-20 was 4.2 Å (Supplementary Fig. 4 and Table 2). Likewise, the lowpass-filtered map of sA3G-VC-RNA-I-20 was used as a picking template for the analyses of sA3G-VC-RNA-II-20, sA3G-VC-RNA20 and sA3G-VC-RNA-IV-20 (Supplementary Fig. 5). The final reconstructions of sA3G-VC-RNA20 (2.8 Å resolution) and sA3G-VC-RNA-IV-20 (2.5 Å resolution) were created including CTF refinement and movie particle polishing. Statistics of datasets and resulting maps are listed in Table 2. The final cryo-EM maps were sharpened using phenix.auto_sharpen implemented in Phenix software[63]. Map examination and visualization were conducted using USCF Chimera[64], Chimera X[65] and PyMol software (Schrödinger, LLC). Electrostatic potentials were calculated with the program, APBS implemented in PyMol.

**Table 2 | Statistics of cryo-EM data collection, refinement and validation**

| Sample<br><br>EMDB deposition<br>PDB deposition | sA3G-VC-RNA-I-20<br><br>EMD35997 | sA3G-VC-RNA-II-20<br><br>EMD-35998 | sA3G-VC- RNA20<br><br>EMD-34412<br>8H0I | sA3G-VC-RNA-IV-20<br><br>EMD-35999<br>8J62 |
|---|---|---|---|---|
| *Data collection and processing* | | | | |
| Microscope | Talos Arctica | Talos Arctica | Titan Krios G1 | Titan Krios G4 |
| Detection camera | Falcon 3EC | Falcon 3EC | Falcon 3EC | Falcon 4 |
| Operating voltage (kV) | 200 | 200 | 300 | 300 |
| Magnification | 120,000 | 120,000 | 96,000 | 215,000 |
| Electron exposure (e/Å²) | 43 | 43 | 37 | 60 |
| Defocus range (µm) | −2.25 to −1.25 | −2.25 to −1.25 | −2.4 to −1.5 | −1.0 to −0.5 |
| Pixel size (Å) | 0.89 | 0.89 | 0.84 | 0.57 |
| Symmetry imposed | C2 | C2 | C2 | C2 |
| Initial particle images (no.) | 1,726,577 | 1,389,140 | 2,831,190 | 2,155,724 |
| Final particle images (no.) | 310,912 | 131,865 | 907,456 | 186,943 |
| Map resolution (Å) (masked) | 4.2 | 3.9 | 2.8 | 2.5 |
| FSC threshold | 0.143 | 0.143 | 0.143 | 0.143 |
| Map resolution range (Å) | 3.93–6.19 | 3.68–5.76 | 2.62–4.86 | 2.37–4.77 |
| *Model refinement* | | | | |
| Initial model used (PDB code) | | | 3IR2/6NIL | 8H0I |
| Model resolution (Å) (masked) | | | 2.9 | 2.7 |
| FSC threshold | | | 0.5 | 0.5 |
| Model sharpening *B* factor (Å²) | | | −123.35 | −70.64 |
| Model composition | | | | |
| Non-hydrogen atoms | | | 15,018 | 10,450 |
| Protein residues | | | 1710 | 1182 |
| Nucleotide bases | | | 38 | 38 |
| Ions | | | 6 | 4 |
| *B* factors (Å) | | | | |
| Protein | | | 79.70 | 52.84 |
| Polynucleotide | | | 62.35 | 54.63 |
| R.m.s. deviations | | | | |
| Bond lengths (Å) | | | 0.002 | 0.002 |
| Bond angles (°) | | | 0.581 | 0.464 |
| Validation | | | | |
| MolProbity score | | | 1.51 | 1.85 |
| Clashscore | | | 7.56 | 9.51 |
| Rotamer outliers (%) | | | 0.67 | 2.91 |
| CaBLAM outliers (%) | | | 1.13 | 0.37 |
| Ramachandran plot | | | | |
| Favored (%) | | | 97.52 | 99.12 |
| Allowed (%) | | | 2.48 | 0.88 |
| Disallowed (%) | | | 0.00 | 0.00 |

## Model building and refinement

Final sharpened maps of sA3G-VC-RNA-20 and sA3G-VC-RNA-IV-20 complexes were used for atomic model building and refinement. Protein coordinate data of A3G CTD (PDB ID 3IR2) and VC (PDB ID 6NIL) were fit to the map using UCSF Chimera[64]. If necessary, model modification, addition, replacement and deletion of amino acids and nucleotides were conducted manually using Coot[66]. Real-space refinements were done using phenix.real_space_refine[67] with non-crystallographic symmetry and secondary structure constraints.

Atomic models were validated using MolProbity[68] and cryo-EM validation tools in Phenix software[69]. Model visualizations were conducted using USCF Chimera[64], Chimera X[65] and PyMol software (Schrödinger, LLC).

## Molecular dynamic calculations

To build the entire model of the ubiquitin ligase complex with A3G, our cryo-EM model (sA3G-VC-RNA-20) and protein coordinate data, PDB IDs 4N9F (VCBC-CUL5)[19], 7B5N (CUL1-Rbx1-ARIH1-NEDD8-E2L2-Ub)[57] and 7ONI (CUL5-Rbx2-ARIH2)[58], were aligned using PyMol (Schrödinger, LLC). If necessary, model modification, addition, replacement and deletion of amino acids were conducted manually. Coordinates of missing amino acids, such as loops, were predicted using Modeller[70]. To relieve local steric hindrances in the resulting model, energy minimization and molecular dynamic calculations (2-fs steps for 1 ns) were conducted with water solvation using NAMD software[71] on a Mac Studio computer with an M1 Ultra processor. Models were examined and visualized using VMD[72] and PyMol (Schrödinger, LLC).

## Statistics and reproducibility

The replicative experiments, such as in vitro ubiquitination assay and Vif-induced degradation assay, were independently performed 3 times ($n$ = 3) for each condition (Fig. 4e, b, d, e and Supplementary Fig. 1f). Otherwise, a representative gel image after a single time experiment is shown for Figs. 1d, 4a, and 5e and Supplementary Figs. 2b–d, f, g and 3e. The observed band densities were used for calculations of averages and standard deviations. The calculated standard deviations were depicted as error bars in figures (Fig. 4c, e and Supplementary Figs. 1f and 10a). In Fig. 4e, statistical tests were conducted to evaluate the statistical significance between two datasets: $F$-test for evaluation of standard deviation equivalency and t-test for assessment of significant difference between two data sets. The significant difference was defined as less than 0.05 of the $p$ value. Source data and uncropped unprocessed scans are provided as a Source data file.

## Reporting summary

Further information on research design is available in the Nature Portfolio Reporting Summary linked to this article.

## Data availability

Atomic coordinates generated in this study for sA3G-VC-RNA20 and sA3G-VC-RNA-IV-20 were deposited in the Protein Data Bank under accession code 8H0I (sA3G-VC-RNA20) and 8J62 (sA3G-VC-RNA-IV-20), respectively. Unsharpened cryo-EM density map, half-maps for sA3G-VC-RNA-I-20, sA3G-VC-RNA-II-20, sA3G-VC-RNA20 and sA3G-VC-RNA-IV-20 were deposited in the Electron Microscopy Data Bank under accession code EMD-35997 (sA3G-VC-RNA-I-20), EMD-35998 (sA3G-VC-RNA-II-20), EMD-34412 (sA3G-VC-RNA20) and EMD-35999 (sA3G-VC-RNA-IV-20), respectively (Table 2). Source data are provided with this paper.

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

## Acknowledgements

This research was supported by the Platform Project for Supporting Drug Discovery and Life Science Research (BINDS) from AMED under grant number JP18am0101076 and by direct funding from OIST (to M.W.); the Toyobo Biotechnology Foundation (to T.K.) and T.K.'s personal funds. H.M. was supported in part by a grant from the U.S. National Institutes of Health R01GM118474/R01AI150478 and federal funds from the National Cancer Institute, National Institutes of Health, under contract 75N91019D00024. We are grateful to Dr. Jeongsik Yong for supervision of T.K. and M.S. during initial RNA cloning. We thank Dr. Chojiro Kojima at Osaka University for providing pCold-GST vectors. We thank Dr. Yohsuke Moriyama and Dr. Keiko Kono at OIST for advice on human cell experiments and western blotting analysis, Dr. Mary Collins and Dr. Melissa Matthews for critical reading of the manuscript, and Dr. Steven D. Aird for technical editing of the manuscript. We thank the OIST Imaging and Analysis Section (IMG) for use of the EM facility, and the OIST Scientific Data Analysis Section (SCDA) for use of the Deigo and Saion high performance computing clusters. We thank Dr. Abhay Kotecha and Dr. Adrian Koh for collecting cryo-EM test data of sA3G-VC-RNA-IV-20 on the Titan Krios G4 Selectris-X at Thermo Fisher Scientific Eindhoven.

## Author contributions

T.K. designed the project strategy. RNA cloning was conducted by T.K. and M.S. T.K. designed sA3G and all wild-type A3G/sA3G mutants, designed all DNA/RNA/hybrid oligomer sequences, cloned all protein constructs used in this study, purified proteins, performed pull-down assays, sA3G-VC-RNA complex preparation, in vitro ubiquitination assays, Vif-induced A3G degradation assays, negative-staining EM experiments, and molecular dynamics calculation. J.H. prepared grapheneoxide-coated cryo-EM grids. S.S. prepared cryo-EM grids. S.S. and M.W. collected cryo-EM data. T.K. processed cryo-EM data with support of S.S., J.H., and T.G.K. T.K. conducted atomic model building and refinement. T.K. and M.W. analyzed molecular structures and recapitulated the entire project. T.K., H.M., and M.W. wrote manuscript. All authors read and contributed to writing the manuscript. M.W. supervised and provided guidance on all cryo-EM experiments, data processing and analyses, and secured funding.

## Competing interests

The authors S.S., J.H., T.G.K., H.S., H.M., and M.W. declare no competing interests. T.K. has filed patents on preparation of the solubility-enhanced human A3G construct, its application for A3G-Vif complex analysis [PCT-JP2019-019938 (2019) and JP patent application number 2021-519919 (2021)]; on DNA/RNA/hybrid-based antagonism against A3G-Vif interaction [PCT-JP2021-42773 (2021)].
