## [Peer Review File · Nature Communications]

Structural insights into RNA bridging between HIV-1 Vif and antiviral factor APOBEC3GEditorial Note: This manuscript has been previously reviewed at another journal that is not operating a transparent peer review scheme. This document only contains reviewer comments and rebuttal letters for versions considered at Nature Communications.

Reviewers' Comments:

Reviewer #2:

Remarks to the Author:

See attached file.

The paper addresses some of the comments from the previous review. However, access to the atomic models pointed out discrepancies between interpretations in the manuscript and the structure. In this reviewer's view, the structures do not support the claim that binding of two Vif proteins to each APOBEC3G (A3G) is relevant to function.

Aided by the atomic models and EM maps, below are comments on the authors' response to specific reviewer points. Original reviewer comments in black, author responses in green, new reviewer comments in red, highlights for quick reading.

Referee #2 (Remarks to the Author):

The human APOBEC3G (A3G) protein potently restricts HIV-1 by binding viral RNA and catalyzing cytosine deamination of the single-stranded DNA product of viral reverse transcription. A long-standing, widely pursued question has been the mechanism by which viral protein Vif recognizes APOBEC3 and recruits an E3 ligase complex to target the restriction factor for proteasomal degradation. The manuscript presents structures from cryo-EM for a 3-protein complex of A3G, HIV-1 Vif, and human CBF β . The structure shows how Vif-CBF β acts as an A3G 'receptor'. The authors found that a single-stranded RNA oligomer stabilized the reconstituted A3G-Vif-CBF β complex. The finding of RNA in the A3G-Vif interface of the structures is significant. Supporting experiments validate that the RNA is relevant to Vif function.

The structure is consistent with published data that RNA binds the N-terminal A3G domain and that this domain also interacts with Vif. Surprisingly, one A3G-RNA bound two copies of Vif-CBF β , i.e. A3G-RNA-(Vif-CBF β)₂. The authors claim biological relevance for this unexpected 1:2 stoichiometry, and cite published data on the importance of specific amino acids in each Vif interface. However, the manuscript lacks **experimental validation for the unexpected 1:2 complex**. This is a critical weakness, particularly as the complex was reconstituted in vitro from purified components, including a defined U-rich RNA 20-mer and A3G and Vif proteins with numerous stabilizing amino acid substitutions. [...]

The main claim of our paper is RNA involvement, not the 1:2 geometry. This is also reflected in the title.

We agree with the reviewer that biological importance of the 1:2 geometry should be carefully examined. In our revised manuscript, we have examined the A3G-Vif assembly produced in human cells with the **co-immunoprecipitation** (co-IP) method (Fig. 1e). The constructs, detection tags and used monoclonal antibodies were carefully chosen, e.g., we confirmed that used anti-hexahistidine antibody didn't show a cross-reaction with A3G nor Vif. Also, we conducted co-IP repeatedly. Our co-IP reproducibly indicated a **ratio of 1.0 : 1.6 (+/-0.2)** for A3G-Vif assembly, respectively (Fig. 1e). It supports that A3G can bind to multiple Vif species in cells although the ratio value was not 1:2 because it is technically impossible to completely inhibit intrinsic RNase activity. Indeed, **addition of excess amount of RNase decreased the ratio to 1:1**.

The co-IP assay is weak validation, especially as only 1:1 stoichiometry survives RNase treatment. See further details below.

We added this stoichiometry data and discussion in the main text and figure 1e.

As we shown in ED Fig. 2 and described in detail in the Supplementary Information, our RNA ligand originated from a co-expression experiment of an A3G variant and Vif in *E. coli*. Although *E. coli* is an artificial host, the RNA sequence (RNA-I-20, ED Fig. 2 and Table I) is a part of the Vif gene, i.e. it corresponds to a viral origin. Even if U-rich, we did not manipulate the original sequence. We are simply proposing that a part of Vif mRNA acts as a molecular glue between A3G and Vif. To obtain the high-resolution structure, we optimized the RNA ligand sequence (e.g., RNA-III-20 and RNA-IV-20). We also compared the structures and showed that A3G-Vif assemblies are essentially identical among the RNA ligands (ED Fig. 6).

For more clarity, we refurbished the main text.

Experimental detail is lacking for “The RNA component was ... subjected to RNA cloning” (lines 1077-1078).

[...] including a defined U-rich RNA 20-mer and A3G and Vif proteins with numerous stabilizing amino acid substitutions. Formation of the complex was weirdly dependent on RNA oligomer sequence and length (extended Fig 3), suggesting that RNA tertiary structure or partial duplex formation may have influenced complex formation. This aspect of the structure should be validated by demonstrating the existence of the 1:2 complex in cells.

Thank you for this important point. We have now carefully performed experiments using co-immunoprecipitation using WT A3G-Vif expressed in human cells, which show a ratio between A3G and Vif of 1.0:1.6 (+/-0.2), and this ratio was decreased to 1:1 in the presence of RNase. This experiment provides strong confidence that our in vitro cryo-EM structure represents the state of the A3G-Vif-RNA complex in living cells.

See the comments above and below.

Additionally, the cryo-EM reconstruction reveals a dimer of the A3G-RNA-(Vif-CBF β)₂. The A3G-A3G contacts mediating dimer formation in [A3G-RNA-(Vif-CBF β)₂]₂ are said to be biologically unclear and were not analyzed. Why dismiss one unexpected interaction and claim biological relevance for the other (without evidence)?

We have now explicitly stated that Vif(blue) may have helped stabilize the complex due to some stabilizing amino acid replacements at its periphery (lines 101-109), and carefully discussed the role of Vif(blue) throughout the revised manuscript.

The “stabilizing amino acid replacements” are at the periphery of sA3G, not Vif (lines 108-109). The following statement on lines 106-107 is misleading and incomplete: “two heteromers of sA3G-VC-RNA20 face each other in a major sA3G-sA3G contact (Fig. 1f).”

In the structure provided by the authors, the sA3G-sA3G “dimer” contact involves only the 61-64 loop of sA3G (ELKY, a non-engineered region). Another, perhaps larger, contact involves sA3G, Vif^{blue}, and RNA – this includes a heavily engineered surface of sA3G. RNA involvement could explain the RNA length & sequence dependence in some assay results.

Thus, Vif^{blue} forms 2 contacts:

- 1) within the “heteromer” that the authors conclude is biologically relevant and
- 2) the “dimer” contact between 2 heteromers that the authors conclude is “non-natural”.

Vif^{blue} stabilizes the “heteromer” and the “dimer” through distinct contacts with 2 sA3G molecules (see figure). As noted, the “dimer” contact is irrelevant because it involves a heavily engineered region of sA3G. The Vif^{blue} “heteromer” contact may be unable to stand on its own without support from the sA3G-Vif^{blue} “dimer” contact. See the comment below for further complications with the Vif^{blue} “heteromer” contact that is claimed to be biologically relevant.

However, no mutations are involved in the A3G-Vif interfaces.

This is not true for the sA3G-Vif^{blue} interface in the “heteromer” contact. In the structure provided by the authors, the engineered sA3G has amino acid substitutions that create a favorable contact or eliminate a charge repulsion:

sA3G has Asp-Pro in place of WT Tyr13-Arg14; the new Asp contacts Vif^{blue} Lys22 (Y13D is missing from ED fig 8a)

sA3G has Gln in place of WT Glu173; the new Gln avoids a repulsion with Vif^{blue} Glu45

The density map indicates that these amino acids are correctly placed. Thus, both of the distinct Vif^{blue} interfaces with sA3G are influenced by stabilizing amino acid replacements in sA3G. I doubt that the “heteromer” contact is relevant.

As mentioned to reviewer 1, our cryo-EM structure and ubiquitination experiments, together with previous biochemical studies by many laboratories suggest that the sA3G-Vif(red) interface is the key interaction for Vif-dependent A3G ubiquitination/degradation.

The recent publications of A3G-Vif structures should be mentioned and cited.

Li *et al.* Emerman, Cheng & Gross (2022) Nature

Ito *et al.* Zhou & Chen (2023) Sci Advances

Aspects of the cryo-EM analysis are alarming. These include 1) repeated auto-picking with a reference structure, especially a reference having enforced C2 symmetry, and 2) re-classification of a particle stack using one class as a reference when several classes had been detected. The Methods section lacks any details that might justify these approaches and, in general, is incomplete. While a negative stain image along with 2D class averages is shown, it would also be helpful to see representative images of the particles in vitrified ice. Euler angle plots should be included, and perhaps also directional FSC plots. The stated resolution of the cryo-EM maps exceeds the appearance of the densities shown in the figures; this is especially clear in the color map of local resolution in the right-most column of extended Fig. 6. None of the density images of high-resolution structures (RNA-III or RNA-IV) in the main text or in extended Figs. 5, 6, 7 or 8 appear to be at the stated resolution of 2.9 – 2.5 Å.

This is an important point. However, the concern about cryo-EM image processing is not an issue.

The initial 3D references were created entirely unbiased from 2D class averages of particles picked with a Gaussian blob. All 3D references have always been low pass filtered to 20A resolution. We will add this statement. Even if some reference bias were present at low resolution, it could not have led to sequence-consistent sidechain information in the final reconstruction, which validates the procedure. The orientation of the C2 symmetry axis (vertical in the paper plane in Fig. 1f – we will add this info) cannot produce duplication of Vifs. Reclassification is common practice for heterogeneous data to maximize the number of particles. Unlike for “Einstein from noise”, our particles are recognizable bona-fide particles in raw images. The reconstruction for RNA-IV-20 complex contains nearly 50% of all original

particles, representing a major subset. We will amend the image processing methods by including Euler angle plots and directional FSC plots as shown in ED Fig. 6. Visual appearance of contoured maps can be misleading and depends on many factors such as contour level, sharpening, post-processing etc. Figures were often rendered at lower contour level (explicitly stated in Fig. legends) to include weaker features prominently. The resolution measurement followed common practice. However, we have now provided full unsharpened and sharpened maps to the editor, so that referees can convince themselves about the quality of our maps.

It is reassuring that a map was the reference for subsequent particle picking (and NOT an atomic model). Nevertheless, a 20-Å low-pass filter is minimal filtering; 60 Å would be more appropriate to insure no bias in the results. The number of particles used in the highest-resolution reconstruction came from <9% of particles originally picked. The extremely terse EM methods section includes no explanation to justify this. It is possible that informative structural classes were left behind.

While the Euler angle plots lack obvious “holes”, all particle stacks exhibited severe preferred orientation.

The best-resolved regions of the maps are lovely! Only 1 result in Table II is deposited in databases, whereas 4 maps should be in the EMDB and 2 models in the PDB.

The C2 symmetry may be justified by the serendipitous formation of a non-natural dimer, facilitated by amino acid replacements in sA3G, but this is not at all clear in the abbreviated description of the EM data processing.

Why was the complex with RNA-III-20, “RNA20”, chosen as the primary structure and used in the figures when the complex with RNA-IV-20 resulted in a higher-resolution map? This choice is not clearly stated or justified. The bar graphs in extended Fig.7 panel j are not helpful. Full plots for each protein and the RNA, as in panel k, would be preferred.

ED Fig. 7 was fully updated for more clarity.

RNA20 produced the most complete map, which was also well-resolved at the periphery. The number of residues built in the RNA20 model was higher than in RNA-IV-20.

The reason for the choice was stated in ED Fig.5 legend;

The final map resolution was 2.5 Å. Although Map (c) [RNA-IV-20] had higher nominal resolution than Map (b) [RNA20] and resolved the core of the structure slightly better, it did not change our interpretation of the RNA density. Because peripheral features were better represented in Map (b), the latter was used for Figure 1f.

The samples (RNA20, RNA-IV-20 and others) differ in their RNA composition and were meant to determine the effect of RNA base replacement. Optimization of resolution was a secondary goal. As such, the maps are not interchangeable.

OK

The negative stain image and six 2D classes of the “U-shaped” ubiquitin ligase complex particles shown in Fig. 5 are not of high enough resolution to see the detail needed to predict organization of the

complex. It was also not clear in the methods if a total of 1,095 particles were picked and then classified into 6 classes, or if the 6 classes shown are the result of classification of a larger negative-stain data set.

We did not claim to determine the structure of the E3 ligase complex; simply evaluated the feasibility of our computationally built model, which was composed of known structures and refined by molecular dynamics. The 6 classes containing 1,095 particles were a dominant subset among 64 classes (total approx. 8,000 particles). We added new panels on this (Fig. 5f-h)

This is *many* classes for few particles! Fig. 5 is unchanged from the previous submission.

The description of the “sA3G” engineered protein is mired in details understandable only to the A3G cognoscenti. A deep dive into references 22 and 23, together with the A3G crystal structures, finally revealed that the A3G “consensus” sequence (extended Fig. 1) pertains to both the N- and C-terminal domains. The apparently 5 engineered substitutions in the A3G C-terminal domain may be L234K, C243A, F310K, C321A and C356A (ref 22) – or not. The full sequence of sA3G should be included. The inclusion of a supplemental figure that clearly shows how each protein in the assembled complex has been modified compared to wild type proteins would be helpful.

The sequences of WT A3G, sA3G NTD and CTD were clearly provided in ED Fig. 1e. As the reviewer points out, there are L234K, C243A, F310K, C321A and C356A mutations on the sA3G CTD, all of which were previously established and are widely used [ref. 22].

“Widely used” refers to APOBEC3 researchers, not the readers of this journal. The engineered sites in the sA3G NTD are clearly presented in ED fig. 1e, but no information is provided for the CTD or the NTD-CTD linker. The lack of a full-length sA3G sequence with clearly indicated amino acid substitutions is an impediment to readers.

The full sequence is provided here:

```
MDPDTFSYNFNNRPILSRRTVWLCYEVERLDNGTWVKMDQHRGQVYSELKYHPEMRFLSLVSKW
KLHRDQEYEV TWYISWSPCTKCARDMATFLQENTHVTLTIFVARLYYFWD PDYQEALRSLAQAGA
TIKIMNYDEFQHCWSKFVYSQGAPFPQWDGLDEYSQALSGMLGEILRHSM DPPTFTFNFNNEPWV
RGRHETYL CYEVERMHNDTWVLLNQRRGFLCNQAPHKHGFLEGRHAELCFLDVI PFWKLDLDQDY
RVTCFTSWSPCFSCAQEMAKFISK NKHVSLCIFTARIYDDQGR CQEGLRTLAEAGAKISIMTYSE
FKHCWDTFVDHQGCPFPQWDGLDEHSQDLSGRLRAILQNQEN
```

The HIV-1 strain that was the source of Vif should be identified.

Thank you. We used a Vif construct based on "pNL4-3 Vif" (originated from Dr. Strebel, NIH). We have now included this information in the methods.

OK

Serendipitous dimer of Vif-sA3G
(Vif-red omitted for clarity)

Reviewer #3:

Remarks to the Author:

In this revision the authors addressed most of the previous comments with solid arguments and data analysis, including the cryo-EM reconstruction procedures. The only major remaining issue is the model of two Vif molecules binding to one sA3G (2:1 stoichiometry). Although the authors have toned down this argument in the revision, it is still presented with strong biological relevance. There are just no convincing functional data to support this. In addition, the recent published wild-type A3G and a mutant rhesus macaque A3G in complex with Vif all showed a 1:1 stoichiometry with a binding mode virtually identical to that of the Vif-red in this study (the published work should not diminish the novelty of this work though, as they were practically done at the same time). I think the authors unnecessarily fell in their own trap by insisting on the interpretation of this rather minor observation. It is common to observe "artificial" oligomer states, as actually shown in the two published A3G-Vif work, or nonnatural stoichiometry in cryo-EM experiments. The key point is that what the authors captured includes the correct, novel RNA-mediated interaction of A3G and Vif. I would suggest the authors further tone down the biological interpretation of the second bound Vif, leaving it as a potential site or artifact from the mutations or cryo-EM conditions. This should not impact the significance of the work and I would recommend publication with this further change.

We appreciate the reviewer's comments which were constructive and helped us to improve our revised paper. Reviewer's comments are orange, and our responses are black. The line numbers refer to the revised manuscript. A comparison highlighting any changes between the original and the new article document is included as additional Word document in the submitted materials.

Referee #3

In this revision the authors addressed most of the previous comments with solid arguments and data analysis, including the cryo-EM reconstruction procedures. The only major remaining issue is the model of two Vif molecules binding to one sA3G (2:1 stoichiometry). Although the authors have toned down this argument in the revision, it is still presented with strong biological relevance. There are just no convincing functional data to support this. In addition, the recent published wild-type A3G and a mutant rhesus macaque A3G in complex with Vif all showed a 1:1 stoichiometry with a binding mode virtually identical to that of the Vif-red in this study (the published work should not diminish the novelty of this work though, as they were practically done at the same time). I think the authors unnecessarily fell in their own trap by insisting on the interpretation of this rather minor observation. It is common to observe "artificial" oligomer states, as actually shown in the two published A3G-Vif work, or nonnatural stoichiometry in cryo-EM experiments. The key point is that what the authors captured includes the correct, novel RNA-mediated interaction of A3G and Vif. I would suggest the authors further tone down the biological interpretation of the second bound Vif, leaving it as a potential site or artifact from the mutations or cryo-EM conditions. This should not impact the significance of the work and I would recommend publication with this further change.

Thank you very much for your constructive critique. We have thoroughly addressed them in this revision. Specifically, we have removed all claims about the biological significance of Vif^{blue} and focused on the RNA-mediated interaction of A3G and Vif. Furthermore, we state explicitly that the engineered mutations may have led to the binding of two Vif molecules and that the second Vif is therefore not biologically relevant. However, these stabilizing interactions may have contributed to our success in obtaining high resolution of the RNA-Vif interactions. These and other points are addressed in detail in the response to reviewer #2 below.

Referee #2

The paper addresses some of the comments from the previous review. However, access to the atomic models pointed out discrepancies between interpretations in the manuscript and the structure. In this reviewer's view, the structures do not support the claim that binding of two Vif proteins to each APOBEC3G

(A3G) is relevant to function.

Thank you for carefully reading our manuscript. In this updated version, we have now resolved earlier discrepancies between the manuscript and our structure, and removed any claims about the biological significance of the second Vif (Vif^{blue}), as follows:

Consistent with current opinions, we made it clear that the sA3G-RNA-Vif^{red} interface is essential for ubiquitination and Vif-induced degradation of sA3G, and that this specific interface is biologically relevant. We now state that “Since Vif^{blue} and CBF β ^{blue} are both involved in this dimer interaction, sA3G:Vif^{blue}CBF β ^{blue}:RNA20 complex formation may be an artifact of these mutations, and the Vif^{blue} contact within each heteromer may be unable to stand on its own without support from the sA3G-Vif^{blue} dimer contact.”

Lines 107-110

The co-IP assay is weak validation, especially as only 1:1 stoichiometry survives RNase treatment. See further details below.

We agree that co-IP assays are weak validation of the stoichiometry of wild-type A3G and Vif interaction in cells. Therefore, we deleted the co-IP part from Fig. 1e and the discussion of A3G-Vif stoichiometry.

Experimental detail is lacking for “The RNA component was ... subjected to RNA cloning” (lines 1077-1078).

This information can be found in the Methods section under “**RNA cloning**”:

“The fraction of sNTD-F126 in complex with VCBC was prepared for RNA extraction with TRIzol, and precipitated with ethanol. The pellet was used for a serial enzymatic reaction and for the cloning:- 3'-end of RNA was extended by poly(A) polymerase (New England BioLabs), 5'-adaptor (5'-rGrUrUrCrArGrArGrUrUrCrUrArCrArGrUrCrCrGrArCrGrArUrC-3') was ligated with T4 RNA ligase (New England BioLabs), followed by reverse transcription with a poly-thymidine DNA oligo, and amplification of the cDNA by polymerase chain reaction (PCR) using *Taq* DNA polymerase (Takara Bio, Japan). Resulting DNAs were inserted into pMD20-T vectors (Takara Bio, Japan), and clones were subjected to DNA sequencing.”

Lines 407-415

The “stabilizing amino acid replacements” are at the periphery of sA3G, not Vif (lines 108-109). The following statement on lines 106-107 is misleading and incomplete:

“two heteromers of sA3G-VC-RNA20 face each other in a major sA3G-sA3G contact (Fig. 1f).”

In the structure provided by the authors, the sA3G-sA3G “dimer” contact involves only the 61-64 loop of sA3G (ELKY, a non-engineered region). Another, perhaps larger, contact involves sA3G, Vif^{blue}, and RNA – this includes a heavily engineered surface of sA3G. RNA involvement could explain the RNA length & sequence dependence in some assay results.

Thus, Vif^{blue} forms 2 contacts:

- 1) within the “heteromer” that the authors conclude is biologically relevant and*
- 2) the “dimer” contact between 2 heteromers that the authors conclude is “non-natural”.*

Vif^{blue} stabilizes the “heteromer” and the “dimer” through distinct contacts with 2 sA3G molecules (see figure). As noted, the “dimer” contact is irrelevant because it involves a heavily engineered region of sA3G. The Vif^{blue} “heteromer” contact may be unable to stand on its own without support from the sA3G-Vif^{blue} “dimer” contact. See the comment below for further complications with the Vif^{blue} “heteromer” contact that is claimed to be biologically relevant.

Thank you again for pointing out this inconsistency clearly and in detail.

We changed the description of the sA3G-sA3G interaction accordingly, resulting in a much more concise interpretation:

“Interestingly, two heteromers of sA3G-VC-RNA20 face each other with their sA3G NTD domains (Fig. 1f), forming a dimer contact between two heteromers. This dimer contact is mediated by sA3G NTD residues (75-SKWKLHRD-83) interacting with Vif^{blue} residues (76-ERDW-79) and CBFβ^{blue} residues (33-RDRP-36) from the other heteromer. The biological relevance of this dimeric arrangement is unclear because preceding residues in wild-type A3G 71-FHWF-74 were all mutated and R78 has been deleted in sA3G (Extended Data Fig. 1e). Since Vif^{blue} and CBFβ^{blue} are involved in both of intra- and inter-heteromeric interactions, sA3G-VC^{blue} formation may be an artifact caused by these mutations and the Vif^{blue} contact within each heteromer may be unable to stand on its own without support from the sA3G-Vif^{blue} dimer contact. On the other hand, the sA3G-VC^{red}-RNA20 interface involves neither mutations nor heterodimer interactions; therefore, it most likely reveals biologically relevant interactions. “

Lines 102-112

This is not true for the sA3G-Vif^{blue} interface in the “heteromer” contact. In the structure provided by the authors, the engineered sA3G has amino acid substitutions that create a favorable contact or eliminate a charge repulsion:

sA3G has Asp-Pro in place of WT Tyr13-Arg14; the new Asp contacts Vif^{blue} Lys22 (Y13D is missing from ED fig 8a)

sA3G has Gln in place of WT Glu173; the new Gln avoids a repulsion with Vif^{blue} Glu45 The density map indicates that these amino acids are correctly placed. Thus, both of the distinct Vif^{blue} interfaces with sA3G are influenced by stabilizing amino acid replacements in sA3G. I doubt that the “heteromer” contact is relevant.

We appreciate the reviewer’s careful observations and investigated those residues in our structure. We agree that Vif^{blue} Lys22 contacts Asp15 of sA3G, but direct contact between Vif^{blue} Lys22 and sA3G Asp13 may be difficult because they are >6 Å apart in the structure. Vif^{blue} Glu45 is also ~6 Å apart from sA3G Gln173 although the local resolution may not be high enough to discuss the detail of side chain arrangements. It has now been clearly stated in the penultimate sentence of the revised paragraph above that the “heteromer” contact may not be able to stand on its own.

Lines 102-112

The recent publications of A3G-Vif structures should be mentioned and cited.

Li et al. Emerman, Cheng & Gross (2023) Nature Ito et al. Zhou & Chen (2023) Sci Advances

These two publications have been mentioned and cited in “Discussion”.

Lines 314-322, references [41] and [57]

It is reassuring that a map was the reference for subsequent particle picking (and NOT an atomic model). Nevertheless, a 20-Å low-pass filter is minimal filtering; 60 Å would be more appropriate to insure no bias in the results. The number of particles used in the highest- resolution reconstruction came from <9% of

particles originally picked. The extremely terse EM methods section includes no explanation to justify this. It is possible that informative structural classes were left behind.

The 2.8 Å resolution structure used for our detailed structural analysis (sA3G-VC-RNA20) was refined using 32% of all particles (907,456 / 2,831,190) after “junk” classes had been eliminated by 2D classification. These ratios are typical and are a testament that the template-based particle selection included a wide variety of image features, including many non-particles.

Upon suggestion of the referee, we reanalyzed our data systematically in two identical experiments, using particle picking templates lowpass-filtered at 20Å and at 60Å. Below is a comparison – 20Å-filtered (**a**, left) and 60Å-filtered template (**b**, right). The 60Å template produced a nearly identical result. While the 60Å reference resulted in even more picked particles, most of them were eliminated in the subsequent 2D classification step (belonging to class sums without any clear protein-like features), leaving a similar number of “good” particles as in the original selection. 3D classification into 3 classes resulted in a single class (class 3) with contiguous protein-like density. The final refined structure from this 3D class was nearly indistinguishable from our previous result (although at 3.1Å at marginally lower resolution due to somewhat smaller number of included particles) and it matches our original structure up to that resolution (FSC at 0.5 threshold for comparing full independent datasets, see FSC plot below). These independent experiments proof that the final structure with sequence-matching molecular features at sidechain resolution could not have arisen as a consequence of reference bias imposed at the step of initial particle selection.

a**b**
While the Euler angle plots lack obvious “holes”, all particle stacks exhibited severe preferred orientation.

Although the Euler angle plot suggests preferred orientations, there were many non-preferred orientations including all possible orientations. We have analyzed the 3D reconstruction with a directional Fourier shell correlation – indeed, our structure shows a reasonable directional FSC towards every axis, suggesting that the additional minority population of non-preferred oriented particles were able to compensate for the observed distribution.

ED Figure 6 b,f,j,n

The best-resolved regions of the maps are lovely! Only 1 result in Table II is deposited in databases, whereas 4 maps should be in the EMDB and 2 models in the PDB.

The C2 symmetry may be justified by the serendipitous formation of a non-natural dimer, facilitated by amino acid replacements in sA3G, but this is not at all clear in the abbreviated description of the EM data processing.

Thank you for this comment! We have now submitted all 4 maps to EMDB and 2 models to the PDB.

Data Availability and Table II

We show multiple examples of refinements with (“w/ C2”) and without (“w/o C2”) symmetry in ED Figs. 4 and 5. Although the C2 symmetry improved the resolution, the particle topology was essentially not affected. The EM data processing section has been updated.

Lines 497-522

*This is *many* classes for few particles! Fig. 5 is unchanged from the previous submission.*

Figure 5 has been changed by removing the computational model based on Vif^{blue} linkage.

The sample was heterogeneous because, as mentioned in the figure caption, the sample consisted of a real reaction mixture of A3G ubiquitination without any chemical fixation or purification process, *i.e.*, a mixture of sA3G, VCBC, NEDDylated CUL5-Rbx2, ARIH2, UBE1, NAE1/UBE3, UBE2F, Ub, E2L3 and RNA20. The goal was to find states of an active, assembled ubiquitin ligase complex, which required a larger number of classes. Although these U-shaped particles provide only a qualitative indication about the complex, our class averages are supported by the recent papers by Li et al., Emerman, Cheng & Gross (Nature 2023), and Ito et al. Zhou & Chen (Sci Advances 2023).

“Widely used” refers to APOBEC3 researchers, not the readers of this journal. The engineered sites in the sA3G NTD are clearly presented in ED fig. 1e, but no information is provided for the CTD or the NTD-CTD linker. The lack of a full-length sA3G sequence with clearly indicated amino acid substitutions is an impediment to readers.

Thank you for this helpful comment.

The entire sA3G sequence including NTD and CTD is now presented in the revised ED Fig. 1e. There is no additional linker or tag between NTD and CTD. For comparison, we also show aligned sequences of wild type human A3G, sNTD, the consensus construct [23], and CTD-2K3A (5 mutations are indicated in red) [22]. The sA3G includes CTD-2K3A as CTD. Matching amino acids between wild-type A3G and sA3G are shaded in red, with mismatches in gray. The figure legend has been updated.

Reviewers' Comments:

Reviewer #3:

Remarks to the Author:

The authors have sufficiently addressed my concerns and the manuscript is suitable for publication.